# A Closer Look at Prototype Classifier for Few-shot Image Classification

**Mingcheng Hou**
The University of Tokyo
hmc1984techjob@gmail.com

**Issei Sato**
The University of Tokyo
sato@g.ecc.u-tokyo.ac.jp

## Abstract

The prototypical network is a prototype classifier based on meta-learning and is widely used for few-shot learning because it classifies unseen examples by constructing class-specific prototypes without adjusting hyper-parameters during meta-testing. Interestingly, recent research has attracted a lot of attention, showing that training a new linear classifier, which does not use a meta-learning algorithm, performs comparably with the prototypical network. However, the training of a new linear classifier requires the retraining of the classifier every time a new class appears. In this paper, we analyze how a prototype classifier works equally well without training a new linear classifier or meta-learning. We experimentally find that directly using the feature vectors, which is extracted by using standard pre-trained models to construct a prototype classifier in meta-testing, does not perform as well as the prototypical network and training new linear classifiers on the feature vectors of pre-trained models. Thus, we derive a novel generalization bound for a prototypical classifier and show that the transformation of a feature vector can improve the performance of prototype classifiers. We experimentally investigate several normalization methods for minimizing the derived bound and find that the same performance can be obtained by using the L2 normalization and minimizing the ratio of the within-class variance to the between-class variance without training a new classifier or meta-learning.

## 1 Introduction

Few-shot learning is used to adapt quickly to new classes with low annotation cost. Meta-learning is a standard training procedure to tackle the few-shot learning problem and the *prototypical network* [1], a.k.a ProtoNet is a widely used meta-learning algorithm for few-shot learning. In ProtoNet, we use a prototype classifier based on meta-learning to predict the classes of unobserved objects by constructing class-specific prototypes without adjusting the hyper-parameters during meta-testing.

ProtoNet has two advantages. (1) Since the nearest neighbor method is applied on query data and class prototypes during the meta-test phase, no hyper-parameters are required in the meta-test phase. (2) The classifiers can quickly adapt to new environments because they do not have to be re-trained for the support set in the meta-test phase when new classes appear. Moreover, the generalization bound of ProtoNet in relation to the number of shots in a support set has been studied [2]. The bound suggests that the performance of ProtoNet depends on the ratio of the within-class variance to the between-class variance of the features extracted using the meta-trained model.

There have been studies on training a new linear classifier on the features extracted using a pre-trained model without meta-learning, which can perform comparably with the meta-learned models [3, 4]. We call this approach the linear-evaluation-based approach. In these studies, the models are trained with the standard classification problem, i.e., models are trained with cross-entropy loss after linear projection from the embedding space to the class-probability space. The linear-evaluation-based

36th Conference on Neural Information Processing Systems (NeurIPS 2022).

approach has three advantages over meta-learning. (1) Training converges faster than meta-learning. (2) Implementation is simpler. (3) Meta-learning decreases in performance if the number of shots does not match between meta-training and meta-testing [2]; however, the linear-evaluation-based approach does not need to take this into account. However, it requires retraining a linear classifier every time a new class appears.

To avoid meta-learning during the training phase and the linear evaluation during the testing phase, we focus on using a prototype classifier in the testing phase and training models in a standard classification manner. As we discuss in section 4, we found that when we directly constructed prototypes from the feature vectors extracted using pre-trained models and applied the nearest neighbor method as in the testing phase in ProtoNet , this does not perform as well as the linear-evaluation-based approach. We hypothesize that the reason is the difference between how the loss is calculated in ProtoNet and pre-trained models. As described in section 3, if we consider a prototype as a pseudo sample average of the features in each class, the loss function of ProtoNet can be considered as having a regularizing effect that makes it closer to the sample average of a class. Since standard classification training computes cross-entropy loss with dot-products to make the features linearly separable, the loss function does not have such an effect and can cause large within-class variance. Figure 1 shows a scatter plot of the features extracted using a neural network with two dimension output trained on CIFAR-10 with ProtoNet(1(a)) and cross-entropy loss with a linear projection layer(1(b)). This figure indicates that the features extracted using a model trained in a standard classification manner get distributed away from the origin and cause large within-class variance along the direction of the class mean vectors, while those of ProtoNet are more centered to its class means. This phenomenon is also observed in face recognition literature [5–8].

We now focus on a theoretical analysis for a prototype classifier. A recent study [2] analyzed an upper bound of risk by using a prototype classifier. The bound depends on the number of shots of a support set, between-class variance, and within-class variance. However, it has two drawbacks. The analysis requires the class-conditioned distribution of features to be Gaussian and to have the same covariance matrix among classes. Moreover, since the bound does not depend on the norm of the feature vectors, it is not clear from the bound what feature-transformation method can lead to performance improvement. Thus, we need to derive a novel bound for a prototype classifier.

Our contributions are threefold.

1. We relax the existing assumption; specifically, the bound does not require that the features be distributed in Gaussian distribution, and each covariance matrix does not have to be the same among classes.

2. We show that our generalization bound consists of three terms: (1) the variance of the norm of feature vectors, (2) the difference in the distribution shape constructed from each class embedding, and (3) the ratio of the within-class variance to the between-class variance.

3. From our theoretical analysis and empirical investigation, we found that reducing two terms is a critical factor to improve the performance of prototype classifiers: the variance of the norm of feature vectors and the ratio of the within-class variance to the between-class variance.

While it is empirically known that the $L2$-normalization of a feature vector and minimize the ratio of the within-class variance to the between-class variance can improve the performance of a prototype classifier, our work is the first to theoretically show that these transformation can improve the performance. The bound of Cao et al. [2] cannot explain the performance improvement of $L2$-normalization because $L2$-normalization does not reduce the ratio of the within-class variance to the between-class variance as shown in Section 4.

A prototype classifier can be applied to any trained feature extractor. It is simpler than linear-evaluation methods such as logistic regression and support vector machine (SVM) for three reasons. (1) A prototype classifier does not require training in the test phase. The training requires time and memory consumption. (2) Since the nearest neighbor method is applied on query data and class prototypes, the computation in the test phase does not depend on the number of classes while linear-evaluation methods do. (3) A prototype classifier requires fewer hyperparameters than linear-evaluation methods. Note that our goal is not to replace linear-evaluation methods with a prototype classifier, but to show that a prototype classifier can be used as a counterpart of the linear-evaluation methods and can be a practically useful first step in tackling few-shot learning problems.

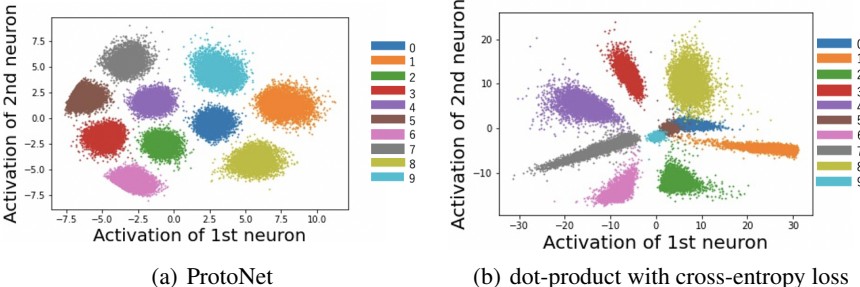

|  | (a) ProtoNet | (b) dot-product with cross-entropy loss |

Figure 1: Distribution of features extracted using a neural network with two dimensional final layer trained on CIFAR-10 with (a): ProtoNet loss and (b): dot-product with cross-entropy loss. The ProtoNet features get distributed closer to its class center than the features extracted using the model trained on cross-entropy loss with linear projection layer.

## 2 Related Work

We summarize related work by describing a prototype classifier with meta-learning, linear-evaluation-based approach without meta-learning, a prototype classifier without meta-learning and theoretical analysis related to the few-shot learning problem.

**Prototype classifier with meta-learning** On the basis of the hypothesis that features well distinguished in the training phase are also useful for classifying new classes, constructing one or multiple prototypes for classifying unseen examples is a widely used approach [1, 9–15]. Certain algorithms compute similarities between multiple prototypes and unseen examples by using their own modules, such as an attention mechanism [9, 14], a relation network [12], reweighting mechanisms by taking into account between-class and within-class interaction [11], and latent clusters [13]. Another line of research is transforming the space of extracted features to a better distinguishable space [16–19] or taking the variance of features into account [20] .

**Update-based meta-learning** In contrast to the prototype classifier with meta-learning, update-based meta-learning adjusts the model parameters to adapt to new classes. Model-agnostic meta-learning (MAML) and its variants [21–23] search for good initialization parameters that adapt to new classes with a few labeled data and a few update steps. These approaches require additional learning of hyper-parameters and training time; thus, they prevent quick adaptation to new classes.

**Linear-evaluation-based approach without meta-learning** Interestingly, recent studies have shown that training a new linear classifier with features extracted using a model trained with cross-entropy loss on base-dataset performs comparably with meta-learning based methods [3, 24]. A more effective method for training a new classifier in few-shot settings has been proposed [25, 26], such as calibrating distribution generated by a support set [25] , self-supervised learning [26], and distilling knowledge to obtain better embeddings [4]. However, similar to update-based meta-learning, these studies require additional hyper-parameters and training for application to new classes; thus if we tackle these points, we would have an alternative method that is easier and more convenient to use.

**A prototype classifier without meta-learing** Recent studies have shown that using a prototype classifier without meta-learning or a linear classifier can lead to comparable performance [24, 27]. Wang et al. [24] has empirically shown that Centering and normalizing the feature vectors can improve the performance of a prototype classifier. Another study empirically shows that applying NCA loss in the training phase can improve the performance of a prototype classifier [27]. However, their study requires a model to be trained with NCA loss while our studies do not require a model to be trained with any specific loss. These studies do not show any theoretical insights on the performance of a prototype classifier and they do not compare with linear-evaluation-based methods.

**Theoretical analysis of few-shot learning** Even though much improvement has been empirically made on few-shot learning, theoretical analysis is scarce. In the context of meta-learning, Du et al.

[28] provided a risk bound on the meta-testing phase that is related to the number of meta-training data and meta-testing data. Lucas et al. [29] derived information-theoretic lower-bounds on min-max rates of convergence for algorithms that are trained on data from multiple sources and tested on novel data. Cao et al. [2] derived a new bound on a prototype classifier and theoretically demonstrated that the mismatch regarding the number of shots in a support set between meta-training and meta-testing degrades the performance of prototypical networks, which has been only experimentally observed. However, their bound depends on two assumptions: the class-conditional distributions of features are Gaussian and have the same covariance matrix among classes. In contrast, we derive a novel bound that does not depend on any specific distribution.

## 3 Theoretical Analysis of Prototype Classifier in Terms of Variance of Norm of Feature Vectors

In this section, we first formulate our problem settings and explain the recent theoretical analysis on a prototype classifier. Next we provide our novel bound for a prototype classifier with the bound related to the variance of the norm of the feature vectors. Finally, we list several methods that can improve the performance of a prototype classifier based on our bound.

### 3.1 Problem Setting

Let $\mathcal{Y}$ be a space of a class, $\tau$ a probability distribution over $\mathcal{Y}$, $\mathcal{X}$ a space of input data, $\mathcal{D}$ a probability distribution over $\mathcal{X}$, $\mathcal{D}_y$ a probability distribution over $\mathcal{X}$ given a class $y$. We define $\mathcal{D}^{\otimes nk}$ and $\mathcal{D}_y^{\otimes n}$ by $\mathcal{D}_y^{\otimes n} = \Pi_{i=1}^n \mathcal{D}_y$ and $\mathcal{D}^{\otimes nk} \equiv \Pi_{i=1}^n \mathcal{D}_i^{\otimes k}$, respectively. We sample $N$ classes from $\tau$ to form the $N$-way classification problem. Denote by $K$ the number of annotated data in each class and $x \in \mathcal{X}, y \in \mathcal{Y}$ as input data and its class respectively. We define a set of support data of class $c$ sampled from $\tau$ as $S_c = \{\boldsymbol{x}_i \mid (\boldsymbol{x}_i, y_i) \in \mathcal{X} \times \mathcal{Y} \cap y_i = c\}_{i=1}^K$ and a set of support data in the $N$-way $K$-shot classification problem as $S = \bigcup_{c=1}^N S_c$. Suppose a feature extractor computes a function $\phi : \mathcal{X} \to \mathbb{R}^D$, where $D$ is the number of the embedding dimensions. $\overline{\phi(S_c)}$ is defined by $\overline{\phi(S_c)} = \frac{1}{K} \sum_{\boldsymbol{x} \in S_c} \phi(\boldsymbol{x})$. Let $\Phi$ be a space of the extractor function $\phi$. Denote by $\mathcal{M} : \Phi \times \mathcal{X} \times (\mathcal{X} \times \mathcal{Y})^{NK} \to \mathbb{R}^N$ a prototype classifier function that computes the probability of input $x$ belonging to class $c$ as follows.

$$\mathcal{M}(\phi, \boldsymbol{x}, S)_c = p_{\mathcal{M}}(y = c | \boldsymbol{x}, S, \phi) = \frac{\exp\left(-\|\phi(\boldsymbol{x}) - \overline{\phi(S_c)}\|^2\right)}{\sum_{l=1}^N \exp\left(-\|\phi(\boldsymbol{x}) - \overline{\phi(S_l)}\|^2\right)}, \tag{1}$$

where $\|v\|^2 = \sum_{d=1}^D \left(v^{(d)}\right)^2$, and $v^{(d)}$ is the $d$-th dimension of vector $v$. The prediction of an input $x$, denoted by $\hat{y} \in \mathcal{Y}$, is computed by taking argmax for $\mathcal{M}(\phi, \boldsymbol{x}, S)$, i.e., $\hat{y} = \operatorname{argmax}\mathcal{M}(\phi, \boldsymbol{x}, S)$. We denote by $\mathbb{E}_{z \sim q(z)}[g(z)]$ an operation to take the expectation of $g(z)$ over $z$ distributed as $q(z)$, and we simply denote $\mathbb{E}_{z \sim q(z)}[g(z)]$ as $\mathbb{E}_z[g(z)]$ when $z$ is obviously distributed on $q(z)$. We define $\operatorname{Var}_{z \sim q(z)}[g(z)]$ as an operation to take the variance of $g(z)$ over $z$ distributed as $q(z)$. With $\mathbb{I}$ denoting the indicator function, we define the expected risk $\mathrm{R}_{\mathcal{M}}$ of a prototype classifier as

$$\mathrm{R}_{\mathcal{M}}(\phi) = \mathbb{E}_{S \sim \mathcal{D}^{\otimes nk}} \mathbb{E}_{c \sim \tau} \mathbb{E}_{\boldsymbol{x} \sim \mathcal{D}_c}[\mathbb{I}[\operatorname{argmax}\mathcal{M}(\phi, \boldsymbol{x}, S) \neq c]. \tag{2}$$

We show a case of multi-class classification in Appendix A.5 due to lack of space. We obtain the bound on multi-class case by Frechet's inequality.

Let $c_1$ and $c_2$ denote any pair of classes sampled from $\tau$. We consider that a query data point $x$ belongs to class $c_1$ and support sets $S$ consist of class $c_1$'s support set and $c_2$'s support set. Then, equation 2 is written as follows.

$$\mathrm{R}_{\mathcal{M}}(\phi) = \mathbb{E}_{S, c_1, \boldsymbol{x}}[\mathbb{I}[\operatorname{argmax}\mathcal{M}(\phi, \boldsymbol{x}, S_{c_1} \cup S_{c_2}) \neq c_1]], \tag{3}$$

$$where \quad \mathbb{E}_{S, c_1, \boldsymbol{x}} = \mathbb{E}_{S \sim \mathcal{D}^{\otimes 2k}} \mathbb{E}_{c_1 \sim \tau} \mathbb{E}_{\boldsymbol{x} \sim \mathcal{D}_{c_1}}.$$

### 3.2 What Feature-Transformation Method Is Expected to Be Effective?

The current theoretical analysis for a prototype classifier [2] has the following two drawbacks (see Appendix A.2 for the details). The first is that the modeling assumption requires a class-conditioned

distribution of the features to follow a Gaussian distribution with the same covariance matrix among classes. For example, when we use the ReLU activation function in the last layer, it is not normally distributed and the class-conditioned distribution does not have the same covariance matrix as shown in Figure 1. The second drawback is that it is not clear what feature-transformation method can reduce the upper bound. A feature-transformation method that maximizes the between-class variance and minimizes the within-class variance such as linear discriminant analysis (LDA) [30] and Embedding Space Transformation (EST) [2] can be expected to improve performance; however, it is not clear how the second term of the denominator changes.

From Figure 1, the distribution of each class feature stretches in the direction of its class-mean feature vector. This property is also observed in metric learning literatures [5, 6]. A model trained in the cross-entropy loss after linear projection from the embedding space to the class-probability space computes a probability of input $x$ belonging to class $c$ with a weight matrix $W \in \mathbb{R}^{D \times K}$ given by

$$p(y = c | \boldsymbol{x}, W, \phi) = \frac{\exp\left(\phi(\boldsymbol{x})^\top W_c\right)}{\sum_{j=1}^N \exp\left(\phi(\boldsymbol{x})^\top W_j\right)}. \tag{4}$$

Comparing equation 4 with equation 1, we found that equation 4 does not have a regularization term similar to the one appearing in equation 1 that forces the features to be close to its class-mean feature vector. This implies that the features extracted using a model trained with the cross-entropy loss using equation 4 are less close to its class-mean feature vector than the ProtoNet loss on equation 1. Through this observation and the property mentioned above, we hypothesize that normalizing the norm of the feature vectors can push the features to each class-mean feature vector and can boost the performance of a prototype classifier trained with cross-entropy loss on equation 4.

### 3.3 Relating Variance of Norm to Upper Bound of Expected Risk

To understand what effect the variance of the norm of the feature vectors has on the performance of a prototype classifier, we analyze how variance contributes to the expected risk when an embedding function $\phi$ is fixed. The following theorem provides a generalization bound for the expected risk of a prototype classifier in terms of the variance of the norm of the feature vectors computed using a feature extractor.

**Theorem 1.** *Let $\mathcal{M}$ be an operation of a prototype classifier for binary classification defined by equation 1. For $\mu_c = \mathbb{E}_{\boldsymbol{x} \sim \mathcal{D}_c}[\phi(\boldsymbol{x})]$, $\Sigma_c = \mathbb{E}_{\boldsymbol{x} \sim \mathcal{D}_c}[(\phi(\boldsymbol{x}) - \mu_c)(\phi(\boldsymbol{x}) - \mu_c)^\top]$, $\mu = \mathbb{E}_{c \sim \tau}[\mu_c]$, $\Sigma = \mathbb{E}_{c \sim \tau}[(\mu_c - \mu)(\mu_c - \mu)^\top]$, and $\Sigma_\tau = \mathbb{E}_{c \sim \tau}[\Sigma_c]$, if $\phi(\boldsymbol{x})$ has the variance of its norm, then the misclassification risk of the prototype classifier for binary classification $\mathrm{R}_\mathcal{M}$ satisfies*

$$\mathrm{R}_\mathcal{M}(\phi) \leq 1 - \frac{4(\mathrm{Tr}(\Sigma))^2}{\mathbb{E}\mathrm{V}[h_{L2}(\phi(\boldsymbol{x}))] + \mathrm{V}_{\mathrm{Tr}}(\Sigma_{c_1}) + \mathrm{V}_{\mathrm{wit}}(\Sigma_\tau, \Sigma, \boldsymbol{\mu}) + \mathbb{E}\,\mathrm{dist}_{\mathrm{L2}}^2(\boldsymbol{\mu}_{c_1}, \boldsymbol{\mu}_{c_2})}, \tag{5}$$

$$where \quad \mathbb{E}\mathrm{V}[h_{L2}(\phi(\boldsymbol{x}))] = \frac{4}{K}\mathbb{E}_{c \sim \tau}\left[\mathrm{Var}_{\boldsymbol{x} \sim \mathcal{D}_c}\left[\|\phi(\boldsymbol{x})\|^2\right]\right], \tag{6}$$

$$\mathrm{V}_{\mathrm{Tr}}(\Sigma_{c_i}) = \left(\frac{4}{K} + \frac{2}{K^2}\right)\mathrm{Var}_{c \sim \tau}\left[\mathrm{Tr}\left(\Sigma_{c_i}\right)\right], \tag{7}$$

$$\mathrm{V}_{\mathrm{wit}}\left(\Sigma_\tau, \Sigma, \boldsymbol{\mu}\right) = \frac{8}{K}\left(\mathrm{Tr}(\Sigma_\tau)\right)\left(\mathrm{Tr}(\Sigma) + \boldsymbol{\mu}^\top\boldsymbol{\mu}\right) + 4\left(\mathrm{Tr}(\Sigma) + \boldsymbol{\mu}^\top\boldsymbol{\mu}\right)^2$$
$$+ 4\mathbb{E}_{c_1, c_2 \sim \tau}\left[\mathrm{Tr}\left(\Sigma_{c_1}\right)\left(\boldsymbol{\mu}_{c_2} - \boldsymbol{\mu}_{c_1}\right)^\top\left(\boldsymbol{\mu}_{c_2} - \boldsymbol{\mu}_{c_1}\right)\right], \tag{8}$$

$$\mathbb{E}\,\mathrm{dist}_{\mathrm{L2}}^2(\boldsymbol{\mu}_{c_1}, \boldsymbol{\mu}_{c_2}) = \mathbb{E}_{c_1, c_2}\left[\left(\left(\boldsymbol{\mu}_{c_1} - \boldsymbol{\mu}_{c_2}\right)^\top\left(\boldsymbol{\mu}_{c_1} - \boldsymbol{\mu}_{c_2}\right)\right)^2\right]. \tag{9}$$

**Remark.** *The term $\mathbb{E}\mathrm{V}[h_{L2}(\phi(\boldsymbol{x}))]$ is the variance of the norm of the feature vectors. The term $\mathrm{V}_{\mathrm{Tr}}(\Sigma_{c_1})$ is the variance of the summation with the diagonal element of the covariance matrix from each class embedding; it can be interpreted as the difference in the distribution shape constructed from each class embedding. The term $\mathrm{V}_{\mathrm{wit}}(\Sigma_\tau, \Sigma, \boldsymbol{\mu})$ is related to the within-class variance. The last term of equation 8 is approximately linear with $\frac{\mathrm{Tr}(\Sigma_{c_1})}{\mathrm{Tr}(\Sigma)}$ because $\frac{(\boldsymbol{\mu}_{c_2} - \boldsymbol{\mu}_{c_1})^\top(\boldsymbol{\mu}_{c_2} - \boldsymbol{\mu}_{c_1})}{\mathrm{Tr}(\Sigma)}$ is supposed to be constant. Thus $\frac{\mathrm{V}_{\mathrm{wit}}(\Sigma_\tau, \Sigma, \boldsymbol{\mu})}{\mathrm{Tr}(\Sigma)}$ is a secondary expression for the ratio of the within-class variance*

to the between-class variance. *The term* $\mathbb{E}\operatorname{dist}^2_{\mathrm{L2}}(\boldsymbol{\mu}_{c_1}, \boldsymbol{\mu}_{c_2})$ *is the expectation of the Euclidean distance between the class-mean vectors .*

This bound has the following properties.

1. Its derivation does not require the features to be distributed in Gaussian distribution and the class-conditioned covariance matrix does not have to be the same among classes

2. The bound can decrease when any of four statistics decreases with fixed between-class variance ($\Sigma$): (i) the variance of the norm of the feature vectors, as discussed in Section 3.2, (ii) the difference in the distribution shape constructed from each class embedding, (iii) the ratio of the within-class variance ($\Sigma_\tau$) to the between-class variance ($\Sigma$), and (iv) the Euclidean distance between the class-mean vectors.

As a result, our bound loosens the modeling assumption of Theorem 2 in Appendix A.2 and has its property. We show the proof in Appendix A.4.

### 3.4 Feature-Transformation Methods

We hypothesize from Theorem 1 that in addition to a feature-transformation method related to equation 6, lowering the ratio of between-class variance to within-class variance can improve the performance of a prototype classifier. As shown in Figure 2 left in experimantal analysis, the value of equation 7 is relatively small compared to equation 6 and equation 8. Regarding equation 9, the ratio of the Euclidean distance between the class mean vectors and the between-class variance is supposed to be constant. Thus, we focus on transformation methods related to equation 6 and equation 8. We analyze four feature-transformation methods: $L_2$-normalization ($L_2$-norm), EST,LDA, EST+$L_2$-norm and LDA+$L_2$-norm. We give the details on each method in Section A.1 due to lack of space.

## 4 Experimental Evaluation

In this section, we experimentally analyzed the effectiveness of the feature-transformation methods mentioned in Section 3.4 under two scenarios: (1) standard object recognition and (2) cross-domain adaptation. The center loss [5] and affinity loss [31] have been proposed to efficiently pull the features of the same class to their centers in the training phase; however,our goal is not to achieve SOTA but, through comparison with existing studies, show that we can achieve the same level of performance without fine-tuning and thus we focus on a widely used pre-trained model in the experiments following the line of studies Chen et al. [3] and Tian et al. [4].

### 4.1 Datasets and Evaluation Protocol

For standard object recognition, we use the *mini*ImageNet dataset, *tiered*ImageNet dataset, the CIFARFS dataset, and the FC100 dataset. **miniImageNet.** The *mini*ImageNet dataset [9] is a standard bench-mark for few-shot learning algorithms in recent studies. It contains 100 classes randomly sampled from ImageNet. Each class contains 600 images. We follow a widely used splitting protocol [32] to split the dataset into $64/16/20$ for training/validation/testing respectively.

**tieredImageNet** The *tiered*ImageNet dataset [33] is another subset of ImageNet but has 608 classes. These classes are grouped into 34 higher level categories in accordance with the ImageNet hierarchy and these categories are split into 20 training (351 classes), 6 validation (97 classes), 8 testing categories (160 classes). This splitting protocol ensures that the training set is distinctive enough from the testing set and makes the problem more realistic than miniImageNet since we generally cannot assume that test classes will be similar to those seen in training.

**CIFAR-FS** The CIFAR-FS dataset [34] is a recently proposed fewshot image classification benchmark, consisting of all 100 classes from CIFAR-100 [35]. The classes are randomly split into 64, 16, and 20 for meta-training, meta-validation, and meta-testing respectively.

**FC100** The FC100 dataset [36] is another dataset derived from CIFAR-100 [35], containing 100 classes which are grouped into 20 categories. These categories are split into 12 categories for training (60 classes), from 4 categories for validation (20 classes), 4 categories for testing (20 classes). This

Table 1: Classification accuracies with ResNet12 on *mini*ImageNet, *tiered*ImageNet, CIFARFS, and FC100 of ProtoNet, linear-evaluation-based methods [3], centering with $L_2$-norm [24], and ours. The Baseline without linear-evaluation methods with accuracy greater than the lower $95\%$ confidence margin of the accuracy of ProtoNet and Baseline are in bold. Regarding to Baseline++, Baseline++ without linear-evaluation methods with accuracy greater than the lower $95\%$ confidence margin of the accuracy of ProtoNet and Baseline++ are in bold. All the methods are our reimplementation .

| | *mini*ImageNet | | *tiered*ImageNet | | CIFAR-FS | | FC100 | |
|---|---|---|---|---|---|---|---|---|
| | 1-shot | 5-shot | 1-shot | 5-shot | 1-shot | 5-shot | 1-shot | 5-shot |
| *methods with meta-learning or linear-evaluation* | | | | | | | | |
| ProtoNet[1] | $53.48 \pm 0.62$ | $73.56 \pm 0.65$ | $55.40 \pm 0.98$ | $77.67 \pm 0.70$ | $63.26 \pm 0.99$ | $79.91 \pm 0.72$ | $35.56 \pm 0.77$ | $51.12 \pm 0.71$ |
| Baseline [3] | $54.54 \pm 0.80$ | $76.50 \pm 0.62$ | $61.67 \pm 0.92$ | $81.62 \pm 0.64$ | $60.62 \pm 0.85$ | $79.79 \pm 0.70$ | $39.72 \pm 0.68$ | $56.04 \pm 0.76$ |
| *A prototype classifier with feature-transformation methods* | | | | | | | | |
| Baseline-w/o-linear , centering+$L_2$-norm[24] | $\mathbf{58.96 \pm 0.83}$ | $\mathbf{75.98 \pm 0.62}$ | $\mathbf{64.88 \pm 0.86}$ | $80.42 \pm 0.64$ | $\mathbf{62.51 \pm 0.86}$ | $\mathbf{79.35 \pm 0.66}$ | $\mathbf{41.58 \pm 0.74}$ | $\mathbf{55.82 \pm 0.76}$ |
| Baseline-w/o-linear | $46.36 \pm 0.58$ | $73.97 \pm 0.62$ | $50.60 \pm 0.87$ | $78.10 \pm 0.67$ | $45.91 \pm 0.89$ | $75.81 \pm 0.79$ | $31.60 \pm 0.61$ | $52.50 \pm 0.74$ |
| Baseline-w/o-linear , $L_2$-norm | $\mathbf{59.60 \pm 1.12}$ | $\mathbf{77.12 \pm 0.44}$ | $\mathbf{64.24 \pm 0.89}$ | $\mathbf{81.93 \pm 0.66}$ | $\mathbf{63.78 \pm 0.87}$ | $\mathbf{80.14 \pm 0.75}$ | $\mathbf{40.34 \pm 0.71}$ | $\mathbf{56.61 \pm 0.71}$ |
| Baseline-w/o-linear , EST | $51.40 \pm 0.83$ | $73.47 \pm 0.62$ | $50.94 \pm 0.93$ | $78.47 \pm 0.68$ | $56.08 \pm 0.94$ | $76.64 \pm 0.76$ | $\mathbf{45.90 \pm 0.85}$ | $\mathbf{64.32 \pm 0.78}$ |
| Baseline-w/o-linear , EST+$L_2$-norm | $\mathbf{57.34 \pm 0.84}$ | $\mathbf{76.90 \pm 0.62}$ | $\mathbf{64.14 \pm 0.91}$ | $\mathbf{81.30 \pm 0.64}$ | $\mathbf{65.19 \pm 0.93}$ | $\mathbf{81.13 \pm 0.69}$ | $\mathbf{49.54 \pm 0.91}$ | $\mathbf{64.12 \pm 0.69}$ |
| Baseline-w/o-linear , LDA | $51.40 \pm 0.83$ | $73.47 \pm 0.62$ | $50.94 \pm 0.93$ | $78.47 \pm 0.68$ | $56.08 \pm 0.94$ | $76.80 \pm 0.76$ | $\mathbf{45.90 \pm 0.85}$ | $\mathbf{64.32 \pm 0.78}$ |
| Baseline-w/o-linear , LDA+$L_2$-norm | $\mathbf{60.29 \pm 0.80}$ | $\mathbf{75.99 \pm 0.66}$ | $\mathbf{64.16 \pm 0.90}$ | $\mathbf{81.96 \pm 0.65}$ | $\mathbf{65.57 \pm 0.89}$ | $\mathbf{80.80 \pm 0.68}$ | $\mathbf{50.97 \pm 0.81}$ | $\mathbf{64.91 \pm 0.76}$ |
| *methods with meta-learning or linear-evaluation* | | | | | | | | |
| Baseline++ [3] | $56.33 \pm 0.81$ | $74.62 \pm 0.60$ | $63.02 \pm 0.91$ | $81.07 \pm 0.69$ | $65.43 \pm 0.95$ | $80.18 \pm 0.75$ | $36.01 \pm 0.64$ | $50.73 \pm 0.72$ |
| *A prototype classifier with feature-transformation methods* | | | | | | | | |
| Baseline++-w/o-linear , centering+$L_2$-norm[24] | $\mathbf{57.50 \pm 0.81}$ | $74.00 \pm 0.60$ | $63.31 \pm 0.91$ | $79.19 \pm 0.68$ | $\mathbf{65.76 \pm 0.90}$ | $79.95 \pm 0.73$ | $\mathbf{37.51 \pm 0.71}$ | $50.38 \pm 0.72$ |
| Baseline++-w/o-linear | $41.18 \pm 0.76$ | $69.48 \pm 0.66$ | $46.52 \pm 0.87$ | $73.85 \pm 0.70$ | $48.28 \pm 0.89$ | $72.79 \pm 0.74$ | $29.72 \pm 0.57$ | $46.85 \pm 0.67$ |
| Baseline++-w/o-linear , $L_2$-norm | $\mathbf{57.96 \pm 0.80}$ | $\mathbf{75.38 \pm 0.61}$ | $\mathbf{65.36 \pm 0.90}$ | $\mathbf{81.08 \pm 0.69}$ | $\mathbf{66.52 \pm 0.92}$ | $\mathbf{80.23 \pm 0.70}$ | $\mathbf{37.16 \pm 0.67}$ | $\mathbf{51.10 \pm 0.71}$ |
| Baseline++-w/o-linear , EST | $47.11 \pm 0.81$ | $69.01 \pm 0.63$ | $52.01 \pm 0.90$ | $68.36 \pm 0.82$ | $52.72 \pm 0.95$ | $72.10 \pm 0.74$ | $\mathbf{37.66 \pm 0.68}$ | $\mathbf{53.34 \pm 0.72}$ |
| Baseline++-w/o-linear , EST+$L_2$-norm | $\mathbf{58.32 \pm 0.81}$ | $\mathbf{75.19 \pm 0.63}$ | $\mathbf{64.51 \pm 0.96}$ | $\mathbf{80.38 \pm 0.70}$ | $\mathbf{66.01 \pm 0.90}$ | $78.78 \pm 0.71$ | $\mathbf{42.98 \pm 0.80}$ | $\mathbf{58.13 \pm 0.72}$ |
| Baseline++-w/o-linear , LDA | $47.42 \pm 0.80$ | $68.43 \pm 0.69$ | $47.76 \pm 0.90$ | $74.41 \pm 0.75$ | $52.19 \pm 0.97$ | $70.91 \pm 0.76$ | $\mathbf{40.40 \pm 0.75}$ | $\mathbf{57.23 \pm 0.75}$ |
| Baseline++-w/o-linear , LDA+$L_2$-norm | $\mathbf{58.26 \pm 0.87}$ | $\mathbf{75.13 \pm 0.66}$ | $\mathbf{65.10 \pm 0.92}$ | $\mathbf{80.39 \pm 0.69}$ | $\mathbf{67.31 \pm 0.96}$ | $\mathbf{79.69 \pm 0.72}$ | $\mathbf{44.83 \pm 0.83}$ | $\mathbf{58.67 \pm 0.74}$ |

Table 2: Classification accuracies with ResNet12 on *mini*ImageNet, *tiered*ImageNet, CIFARFS, and FC100 of methods in current studies and ours. The best performing methods and any other runs within 95% confidence margin are in bold. Methods with † show the results of our reimplementation following Section 4.2

| | *mini*ImageNet | | *tiered*ImageNet | | CIFAR-FS | | FC100 | |
|---|---|---|---|---|---|---|---|---|
| | 1-shot | 5-shot | 1-shot | 5-shot | 1-shot | 5-shot | 1-shot | 5-shot |
| *methods with meta-learning or linear-evaluation* | | | | | | | | |
| MetaOptNet [36]† | $59.65 \pm 0.60$ | $75.53 \pm 0.62$ | $63.01 \pm 0.72$ | $80.01 \pm 0.64$ | $66.03 \pm 0.77$ | $79.80 \pm 0.54$ | $40.1 \pm 0.63$ | $53.3 \pm 0.62$ |
| TapNet [17] | $\mathbf{61.65 \pm 0.15}$ | $76.36 \pm 0.10$ | $63.08 \pm 0.15$ | $80.26 \pm 0.12$ | - | - | - | - |
| *A prototype classifier with feature-transformation methods* | | | | | | | | |
| Baseline-w/o-linear , $L_2$-norm | $59.60 \pm 1.12$ | $\mathbf{77.12 \pm 0.44}$ | $64.24 \pm 0.89$ | $\mathbf{81.93 \pm 0.66}$ | $63.78 \pm 0.87$ | $80.14 \pm 0.75$ | $40.34 \pm 0.71$ | $56.61 \pm 0.71$ |
| Baseline-w/o-linear , EST+$L_2$-norm | $57.34 \pm 0.84$ | $\mathbf{76.90 \pm 0.62}$ | $64.14 \pm 0.91$ | $81.30 \pm 0.64$ | $65.19 \pm 0.93$ | $\mathbf{81.13 \pm 0.69}$ | $49.54 \pm 0.91$ | $\mathbf{64.12 \pm 0.69}$ |
| Baseline-w/o-linear , LDA+$L_2$-norm | $60.29 \pm 0.80$ | $75.99 \pm 0.66$ | $64.16 \pm 0.90$ | $\mathbf{81.96 \pm 0.65}$ | $65.57 \pm 0.89$ | $80.80 \pm 0.68$ | $50.97 \pm 0.81$ | $\mathbf{64.91 \pm 0.76}$ |
| Baseline++-w/o-linear , $L_2$-norm | $57.96 \pm 0.80$ | $75.38 \pm 0.61$ | $\mathbf{65.36 \pm 0.90}$ | $81.08 \pm 0.69$ | $66.52 \pm 0.92$ | $80.23 \pm 0.70$ | $37.16 \pm 0.67$ | $51.10 \pm 0.71$ |
| Baseline++-w/o-linear , EST+$L_2$-norm | $58.32 \pm 0.81$ | $75.19 \pm 0.63$ | $64.51 \pm 0.96$ | $80.38 \pm 0.70$ | $66.01 \pm 0.90$ | $78.78 \pm 0.71$ | $42.98 \pm 0.80$ | $58.13 \pm 0.72$ |
| Baseline++-w/o-linear , LDA+$L_2$-norm | $58.26 \pm 0.87$ | $75.13 \pm 0.66$ | $\mathbf{65.10 \pm 0.92}$ | $80.39 \pm 0.69$ | $\mathbf{67.31 \pm 0.96}$ | $79.69 \pm 0.72$ | $44.83 \pm 0.83$ | $58.67 \pm 0.74$ |

splitting protocol is similar to the protocol used in tieredImageNet so that the training set is distinctive enough from the testing set and makes the problem more realistic than CIFAR-FS.

For cross-domain scenario, we use the *mini*ImageNet dataset during pre-training stage and we use the CUB-200-2011 dataset [37], a.k.a CUB during testing stage.

**CUB** The CUB dataset contains 200 classes and 11,788 images in total. Following the evaluation protocol of Chen et al. [3], we split the dataset into $64/16/20$ for training/validation/testing respectively.

### 4.2 Implementation Details

We compared the feature-transformation methods against ProtoNet [1], Baseline, and Baseline++ [3]. For Baseline and Baseline++, we trained the linear projection layer on a support set. The difference between Baseline and Baseline++ is that the norm of the linear projection layer and features in Baseline++ are normalized to be constant. We call Baseline and Baseline++ as "linear evaluation" methods. We also compared with the feature-transformation method proposed in a previous study [24]. In that study they transformed the features so that the mean of all features was the origin before $L_2$-normalization and then used a prototype classifier without training a new linear classifier. We call this operation centering+$L_2$-norm. We re-implemented these methods following the training procedure in a previous study [3]. In the pre-training stage, where the cross-entropy loss was used and meta-learning was not used, we trained $400$ epochs with a batch size of $16$. In the meta-training stage for ProtoNet, we trained $60,000$ episodes for 1-shot and $40,000$ episodes for 5-shot tasks. We

used the validation set to select the training episodes with the best accuracy. We evaluated the model every 5 epochs, not every epoch as experimented in Chen et al. [3] , and we performed early stopping when the validation score does not improve for 50 epochs. In each episode, we sampled $N$ classes to form $N$-way classification (in meta-training $N$=20 and meta-testing $N$=5 following the original study for ProtoNet [1]). For each class, we selected $K$ labeled instances as our support set and 16 instances for the query set for a $K$-shot task.

In the linear evaluation or meta-testing stage for all methods, we averaged the results over 600 trials. In each trial, we randomly sampled 5 classes from novel classes, and in each class, we also selected $K$ instances for the support set and 16 for the query set. For Baseline and Baseline++, we used the entire support set to train a new classifier for 100 iterations with a batch size of 4. With ProtoNet, we used the models trained in the same shots as meta-testing stage since a mismatch in the number of shots between meta-training and meta-testing degrades performance [2]. All methods were trained from scratch, and the Adam optimizer with an initial learning rate of $10^{-3}$ was used. We applied standard data augmentation including random crop, horizontal-flip, and color jitter in both the training stages. We used a ResNet12 network and ResNet18 network following the previous study [3, 17].

For LDA we set $\lambda = 0.0001$ for equation 11 in Appendix A.1 and for EST we set the dimensions of the projected space to 60 following the settings of the original study [2]. Following the procedure of EST, we calculate the transformation matrix of LDA based on the features of the training dataset.

## 4.3 Results

We present the experimental results of standard object recognition in Table 1 on the basis of backbones with ResNet12 for a comprehensive comparison. We show the result of backbones with ResNet18 in Table 5. We present the discussion on cross-dataset scenario and the result of the scenario in 4.5.

**Comparison of feature-transformation methods with ProtoNet, linear evaluation methods, and centering+$L_2$-norm**    From Table 1, we can observe that the prototype classifier with $L_2$-norm, EST+$L_2$-norm, LDA+$L_2$-norm performs comparably with ProtoNet and the linear-evaluation-based approach in all settings. Comparing the feature-transformation methods described in Section 3.4 with centering+$L_2$-norm, centering+$L_2$-norm can slightly improve the performance of the prototype classifier in several 1-shot settings . However, in 5-shot settings, the boost decreases and even performs worse than linear-evaluation methods, e.g. *mini*ImageNet and *tiered*ImageNet.

**Comparison among feature-transformation methods**    In the 1-shot setting, although EST falls short of ProtoNet and the linear-evaluation-based approach, it also improves the performance of a prototype classifier. The performance gain of both $L_2$-norm, EST, LDA, EST+$L_2$-norm and LDA + $L_2$-norm decrease when the number of shots increases. This phenomenon can be explained through Theorem 1. The term relating to the variance of the norm and the ratio of the within-class variance to the between-class variance depends on $K$. Since the term diminishes as $K$ increases, the performance gain of the feature-transformation methods decreases.

**Comparison of feature-transformation methods with current studies**    Table 2 shows the performance of MetaOptNet [36], TapNet [17] and feature-transformation methods that performs comparable with ProtoNet or linear-evaluation-based methods in Table 1. We can observe that a prototype classifier can achieve comparable performance with current studies in most of the settings with the feature transformations. Moreover, we can further boost the performance of a prototype classifier by combining $L_2$-norm and the methods to reduce the ratio of the within-class variance to the between-class variance, e.g. CIFAR-FS and FC100.

**Discussion on feature-transformation methods**    LDA and EST in the 5-shot setting do not improve the performance so much compared with $L_2$-norm variants while in the 1-shot settings, LDA and EST improve the performance. This is because, in the 5-shot settings, the values computed from equation 8 get smaller with larger $K$, and the effect of equation 8 on the risk of a prototype classifier in the 1-shot settings is larger. Especially, EST and LDA outperform $L_2$-norm in FC100. We found from Figure 2 that the features of FC100 show the largest ratio of the within-class to between-class variance among all datasets. Thus the method of reducing the ratio works better with FC100 features than with any other dataset's features. As discussed in Section 4.3, the combination of feature-transformation

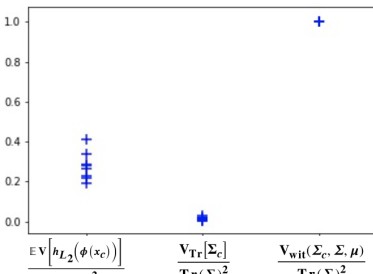 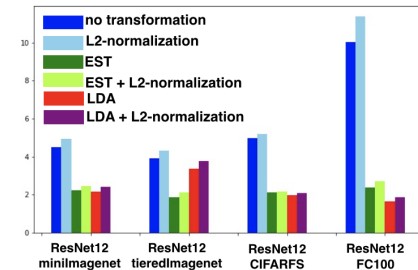

Figure 2: Left: We plotted the values shown in equations 6, 7, 8 divided by $\mathrm{Tr}(\Sigma)^2$. The values are computed from test-split of each dataset with ResNet12 and ResNet18. We scaled the values so that each dataset's $\frac{\mathrm{V_{wit}}(\Sigma_\tau, \Sigma, \mu)}{\mathrm{Tr}(\Sigma)^2}$ to be 1 for simplicity. Right: We plotted the ratio of the within-class variance to the between-class variance ($\frac{\mathrm{Tr}(\Sigma_\tau)}{\mathrm{Tr}(\Sigma)}$) before and after applying $L_2$-norm, EST, LDA, EST + $L_2$-norm and LDA+$L_2$-norm of each dataset.

methods that reduce the variance of the norm and the ratio of the within-class variance to the between-class variance can further boost the performance of a prototype classifier. This indicates that reducing the term equation 6 and equation 8 is an important factor for the performance of a prototype classifier.

## 4.4 The comparison of the ratio of the within-class variance to the between-class variance before and after applying $L_2$norm, EST and LDA

We show the ratio of the within-class variance to the between-class variance before and after applying $L_2$-norm, EST, LDA, EST + $L_2$-norm, LDA + $L_2$-norm in Figure 2 right. We calculated the ratio of the between-class variance to each class's variance of each dataset. This figure shows that $L_2$-normalization slightly changes the ratio while EST and LDA can reduce the ratio. The combination of EST and $L_2$-norm or LDA and $L_2$-norm can reduce the ratio while decreasing the variance of the norm of feature vectors to 0. Therefore, EST + $L_2$-norm and LDA + $L_2$-norm performs better than $L_2$-norm. Moreover, $L_2$-norm does not affect the bound of Cao et al. [2] because their bound depends on the ratio of the between-class variance to each class's variance as shown in 2. Thus the bound of Cao et al. [2] cannot explain the performance improvement of a prototype classifier with $L_2$-norm while our bound can.

## 4.5 Performance results of cross dataset

We show in Table 3 the performance results of cross-dataset scenario in this section with 95% confidence margin. From the table, we can observe that the prototype classifier with $L_2$-norm, EST+$L_2$-norm and LDA+$L_2$-norm performs comparably with linear-evaluation-based methods. Therefore, our analysis results still hold on the cross-domain scenario.

We also found that ProtoNet performs worse than linear-evaluation-based methods and a prototype classifier with feature-transformation methods. In this scenario, we should focus on improving the performance of a prototype classifier with features extracted from a model trained without meta-learning, e.g, Baseline and Baseline++. Since there is no guarantee that the features of a model trained with cross-entropy loss will be distributed in Gaussian distribution, Cao et al. [2]'s bound cannot explain the performance improvement of the prototype classifier while our analysis can.

## 4.6 The gap between the performance of a linear classifier and a prototype classifier

Since we hypothesize that the performance gap between protototype classifiers and ProtoNet is related to the difference of how the loss function is calculated, we experimentally analyzed that how the gap changes when cosince distance or innerproduct is used in ProtoNet instead of Euclidean distance. We show in Table 4 the performance of ProtoNet with cosine similarities and innerproduct.

The performance gap between a prototype classifier based on the features of Baseline and ProtoNet with cosine distance or inner product is decreased compared to the gap between a prototype classifier based on the features of Baseline and a ProtoNet with Euclidean distance.

Table 3: Classification accuracies with ResNet12 and ResNet18 on cross-domain scenario (*mini*ImageNet → CUB) of ProtoNet, linear-evaluation-based methods [3], centering with $L_2$-norm [24], and ours. The notation is same as Table 1.

| | *mini*ImageNet → CUB ResNet12 | | *mini*ImageNet → CUB ResNet18 | |
|---|---|---|---|---|
| | **1-shot** | **5-shot** | **1-shot** | **5-shot** |
| *methods with meta-learning or linear-evaluation* | | | | |
| ProtoNet[1] | $38.47 \pm 0.69$ | $61.47 \pm 0.68$ | $44.01 \pm 0.79$ | $61.79 \pm 0.75$ |
| Baseline [3] | $46.91 \pm 0.81$ | $66.55 \pm 0.72$ | $46.36 \pm 0.78$ | $66.97 \pm 0.73$ |
| *A prototype classifier with feature-transformation methods* | | | | |
| Baseline-w/o-linear , centering+$L_2$-norm[24] | $\mathbf{46.70 \pm 0.79}$ | $65.36 \pm 0.70$ | $\mathbf{46.56 \pm 0.77}$ | $65.96 \pm 0.72$ |
| Baseline-w/o-linear | $43.05 \pm 0.72$ | $63.47 \pm 0.67$ | $42.11 \pm 0.74$ | $64.31 \pm 0.73$ |
| Baseline-w/o-linear , $L_2$-norm | $\mathbf{47.64 \pm 0.79}$ | $\mathbf{66.31 \pm 0.73}$ | $46.55 \pm 0.83$ | $\mathbf{67.21 \pm 0.73}$ |
| Baseline-w/o-linear , EST | $43.27 \pm 0.77$ | $61.59 \pm 0.70$ | $43.90 \pm 0.77$ | $63.50 \pm 0.72$ |
| Baseline-w/o-linear , EST+$L_2$-norm | $\mathbf{46.33 \pm 0.78}$ | $\mathbf{66.28 \pm 0.71}$ | $47.10 \pm 0.83$ | $\mathbf{66.33 \pm 0.74}$ |
| Baseline-w/o-linear , LDA | $43.22 \pm 0.74$ | $62.47 \pm 0.71$ | $44.02 \pm 0.74$ | $65.34 \pm 0.75$ |
| Baseline-w/o-linear , LDA+$L_2$-norm | $\mathbf{47.65 \pm 0.80}$ | $\mathbf{66.94 \pm 0.72}$ | $48.47 \pm 0.82$ | $\mathbf{67.21 \pm 0.69}$ |
| *methods with meta-learning or linear-evaluation* | | | | |
| Baseline++ [3] | $45.56 \pm 0.82$ | $66.00 \pm 0.74$ | $46.26 \pm 0.87$ | $63.53 \pm 0.71$ |
| *A prototype classifier with feature-transformation methods* | | | | |
| Baseline++-w/o-linear , centering+$L_2$-norm[24] | $\mathbf{47.36 \pm 0.82}$ | $64.44 \pm 0.75$ | $\mathbf{46.55 \pm 0.79}$ | $64.52 \pm 0.71$ |
| Baseline++-w/o-linear | $40.68 \pm 0.71$ | $61.02 \pm 0.71$ | $39.72 \pm 0.70$ | $60.00 \pm 0.74$ |
| Baseline++-w/o-linear , $L_2$-norm | $\mathbf{47.03 \pm 0.83}$ | $\mathbf{65.86 \pm 0.75}$ | $\mathbf{46.87 \pm 0.82}$ | $\mathbf{65.39 \pm 0.71}$ |
| Baseline++-w/o-linear , EST | $40.28 \pm 0.73$ | $56.79 \pm 0.72$ | $40.11 \pm 0.74$ | $54.95 \pm 0.73$ |
| Baseline++-w/o-linear , EST+$L_2$-norm | $\mathbf{46.64 \pm 0.84}$ | $\mathbf{65.39 \pm 0.74}$ | $\mathbf{45.96 \pm 0.78}$ | $\mathbf{64.23 \pm 0.77}$ |
| Baseline++-w/o-linear , LDA | $40.44 \pm 0.75$ | $57.55 \pm 0.69$ | $39.54 \pm 0.72$ | $56.51 \pm 0.78$ |
| Baseline++-w/o-linear , LDA+$L_2$-norm | $\mathbf{46.32 \pm 0.81}$ | $\mathbf{65.35 \pm 0.76}$ | $\mathbf{45.60 \pm 0.81}$ | $\mathbf{64.17 \pm 0.73}$ |

Table 4: Classification accuracies with ResNet12 on *mini*ImageNet and *tiered*ImageNet of ProtoNet with feature-transformation-methods.

| | *mini*ImageNet | | *tiered*ImageNet | |
|---|---|---|---|---|
| | **1-shot** | **5-shot** | **1-shot** | **5-shot** |
| ProtoNet[1] | $54.42 \pm 0.86$ | $73.56 \pm 0.68$ | $56.96 \pm 0.98$ | $78.38 \pm 0.71$ |
| ProtoNet-w-cosine similarity | $53.98 \pm 0.83$ | $72.38 \pm 0.66$ | $53.17 \pm 0.91$ | $73.61 \pm 0.77$ |
| ProtoNet-w-innerproduct | $53.87 \pm 0.83$ | $71.86 \pm 0.69$ | $49.10 \pm 1.00$ | $70.11 \pm 0.84$ |
| Baseline-w/o-linear | $46.36 \pm 0.58$ | $73.97 \pm 0.62$ | $50.60 \pm 0.87$ | $78.10 \pm 0.67$ |

## 5 Conclusion

We theoretically and experimentally analyzed how the variance of the norm of feature vectors affects the performance of a prototype classifier. We derived a generalization bound that does not require the features to be distributed in Gaussian distribution and the class-conditioned covariance matrix does not have to be the same among classes. We found that using EST+$L_2$-norm makes the classifier comparable with ProtoNet and the linear-evaluation-based approach. Our experiments show that when the number of shots in a support set increases, the performance gain from a feature-transformation method decreases, which is consistent with the results of theoretical analysis.

One limitation of this paper is that, in the first place, the prototype classifier works well under the assumption that the data distribution for each class is unimodal and isotropic modeling. However, we consider that this assumption may be sound in practice because the multimodality of the data distribution in a class is typically caused by the way the data is annotated. Since the number of a support set is small in few-shot settings (for example, in 5-way 5-shot, the number of labeled data is 25), we can easily check the labels and re-label them so that they don't become multi-modal.

**Broader impact.** We believe that our work shows that a prototypical classifier is expected to be a practically useful first step in tackling few-shot learning problems because of its simplicity. The progress in few-shot learning can impact importnant problems such as medical images and re-identification problems. We also recognize our work might constitute a threat that authoritarian entities deploy few-shot learning algorithms for surveillance.

## 6 Acknowledgments

This work was supported by JSPS KAKENHI Grant Number 20H04239 Japan.

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
