# A Appendix

## A.1 Detail of Feature-Transforming Methods

$L_2$ **normalization ($L_2$-norm)** From equation 6, normalizing the norm of the feature vectors can improve the performance of a prototype classifier. We denote a function normalizing the norm as $\psi_{L2}$ given by

$$\psi_{L2}(\phi(\boldsymbol{x})) = \frac{\phi(\boldsymbol{x})}{\|\phi(\boldsymbol{x})\|}. \tag{10}$$

**LDA** From equation 8, decreasing the ratio of the within-class variance to the between-class variance can improve the performance of a prototype classifier. LDA [30] is widely used to search for a projection space that maximizes the between-class variance and minimizes the within-class variance. It computes the eigenvectors of a matrix $\hat{\Sigma}_\tau^{-1}\hat{\Sigma}$, where $\hat{\Sigma}$ is the covariance matrix of prototypes and $\hat{\Sigma}_\tau$ is the class-conditioned covariance matrix. Since the number of data is small in few-shot settings, $\hat{\Sigma}_\tau^{-1}$ cannot be estimated stably and we add a regularizer term to $\hat{\Sigma}_\tau$ and define it as $\hat{\Sigma}_{\tau\text{reg}}$.

$$\hat{\Sigma}_{\tau\text{reg}} = \hat{\Sigma}_\tau + \lambda I, \tag{11}$$

where $\lambda \in R^1, I \in R^{D \times D}$ is identity matrix.

**EST** Since computation of LDA is unstable, we also analyze the effect of EST [2]. EST computes eigenvectors of a matrix $\hat{\Sigma} - \rho\hat{\Sigma}_\tau$: the difference between the covariance matrix of the class mean vectors and the class mean covariance matrix with weight parameter $\rho$. Similar to LDA, EST also searches for the projection space that maximizes the $\Sigma$ and minimizes the $\Sigma_\tau$.

**EST+$L_2$-norm** We hypothesize that the combination of the transforming methods can improve the performance of a prototype classifier independently from each other. Specifically, we focus on reducing equation 6 and equation 8 by combining **EST** and $L_2$-**norm**. We first apply EST to reduce equation 8 and after that we apply $L_2$-norm to reduce equation 6. At the end of the operation we want the variance of the norm to be 0, thus we apply EST and after that we apply $L_2$-norm.

**LDA+$L_2$-norm** We focus on reducing equation 8 and equation 7 by combining **LDA** and $L_2$-**norm**. Following the similar procedure of **EST+$L_2$-norm**, we first apply LDA and after that we apply $L_2$-norm.

## A.2 Existing Upper Bound on Expected Risk for Prototype Classifier

To analyze the behavior of a prototype classifier, we start from the current study [2]. The following theorem is the upper bound of the expected risk of prototypical networks with the next conditions.

- The probability distribution of an extracted feature $\phi(\boldsymbol{x})$ given its class $y = c$ is Gaussian i.e $\mathcal{D}_y = \mathcal{N}(\mu_c, \Sigma_c)$, where $\mu_c = \mathbb{E}_{\boldsymbol{x} \sim \mathcal{D}_c}[\phi(\boldsymbol{x})]$ and $\Sigma_c = \mathbb{E}_{\boldsymbol{x} \sim \mathcal{D}_c}[(\phi(\boldsymbol{x}) - \mu_c)(\phi(\boldsymbol{x}) - \mu_c)^\top]$.
- All class-conditioned distributions have the same covariance matrix, i.e., $\forall (c, c'), \Sigma_c = \Sigma_{c'}$.

**Theorem 2** (Cao et al. [2]). *Let $\mathcal{M}$ be an operation of a prototype classifier on binary classification defined by equation 1. Then for $\mu = \mathbb{E}_{c \sim \tau}[\mu_c]$ and $\Sigma = \mathbb{E}_{c \sim \tau}[(\mu_c - \mu)(\mu_c - \mu)^\top]$, the misclassification risk of the prototype classifier on binary classification $\mathrm{R}_\mathcal{M}$ satisfies*

$$\mathrm{R}_\mathcal{M}(\phi) \leq 1 - \frac{4\,\mathrm{Tr}(\Sigma)^2}{8(1 + 1/k)^2\,\mathrm{Tr}\left(\Sigma_c^2\right) + 16(1 + 1/k)\,\mathrm{Tr}\left(\Sigma\Sigma_c\right) + \mathbb{E}\,\mathrm{dist}_{\mathrm{L2}}^2(\boldsymbol{\mu}_{c_1}, \boldsymbol{\mu}_{c_2})}, \tag{12}$$

*where* $\mathbb{E}\,\mathrm{dist}_{\mathrm{L2}}^2(\boldsymbol{\mu}_{c_1}, \boldsymbol{\mu}_{c_2}) = \mathbb{E}_{c_1, c_2}\left[((\boldsymbol{\mu}_{c_1} - \boldsymbol{\mu}_{c_2})^\top(\boldsymbol{\mu}_{c_1} - \boldsymbol{\mu}_{c_2}))^2\right]$.

We show the detail of the derivation in Appendix A.3.

### A.3 Reviewing Derivation Details of Theorem 2 (Cao et al. [2])

We briefly review the derivation of Theorem 2. In prototype classifier, from equation 1 and equation 3, $R_{\mathcal{M}}$ is written with sigmoid function $\sigma$ as follows:

$$
\begin{aligned}
R_{\mathcal{M}}(\phi) =& \Pr_{c_1,c_2\sim\tau,\boldsymbol{x}\sim\mathcal{D}_{c_1},S\sim\mathcal{D}\otimes 2K}\left(\sigma(\|\phi(\boldsymbol{x})-\overline{\phi(S_{c_2})}\|-\|\phi(\boldsymbol{x})-\overline{\phi(S_{c_1})}\|)\leq\frac{1}{2}\right)\\
=& \Pr(\alpha<0),
\end{aligned}
\tag{13}
$$

where $\alpha \triangleq \|\phi(\boldsymbol{x})-\overline{\phi(S_{c_2})}\|-\|\phi(\boldsymbol{x})-\overline{\phi(S_{c_1})}\|$. we bound equation 13 with expectation and variance of $\alpha$ by following proposition.

**Proposition 1.** *From the one-sided Chebyshev's inequality, it immediately follows that:*

$$
\mathrm{R}_{\mathcal{M}}(\phi) = \Pr(\alpha<0) \leq 1 - \frac{\mathbb{E}_{S\sim\mathcal{D}\otimes 2k}\mathbb{E}_{c_1,c_2\sim\tau}\mathbb{E}_{\boldsymbol{x}\sim\mathcal{D}_{c_1}}[\alpha]^2}{\mathrm{Var}_{S,c_1,c_2,\boldsymbol{x}}[\alpha]+\mathbb{E}_{S\sim\mathcal{D}\otimes 2k}\mathbb{E}_{c_1,c_2\sim\tau}\mathbb{E}_{\boldsymbol{x}\sim\mathcal{D}_{c_1}}[\alpha]^2}.
\tag{14}
$$

equation 13 can be further write down as follows by Law of Total Expectation

$$
\begin{aligned}
\mathrm{Var}_{S,c,\boldsymbol{x}}(\alpha) &= \mathbb{E}_{S\sim\mathcal{D}\otimes 2k}\mathbb{E}_{c_1,c_2\sim\tau}\mathbb{E}_{\boldsymbol{x}\sim\mathcal{D}_c}[\alpha^2] - \left(\mathbb{E}_{S\sim\mathcal{D}\otimes 2k}\mathbb{E}_{c_1,c_2\sim\tau}\mathbb{E}_{\boldsymbol{x}\sim\mathcal{D}_{c_1}}[\alpha]\right)^2\\
&= \mathbb{E}_{c_1,c_2}\mathbb{E}_{\boldsymbol{x},S}[\alpha^2|c_1,c_2] - \left(\mathbb{E}_{S\sim\mathcal{D}\otimes 2k}\mathbb{E}_{c_1,c_2\sim\tau}\mathbb{E}_{\boldsymbol{x}\sim\mathcal{D}_{c_1}}[\alpha]\right)^2\\
&= \mathbb{E}_{c_1,c_2}[\mathrm{Var}_{\boldsymbol{x},S}(\alpha|c_1,c_2)+\mathbb{E}_{\boldsymbol{x},S}[\alpha|c_1,c_2]^2] - \mathbb{E}_{c_1,c_2\sim\tau}\mathbb{E}_{\boldsymbol{x}\sim\mathcal{D}_{c_1}}\mathbb{E}_{S\sim\mathcal{D}_{\otimes 2K}}[\alpha]^2.
\end{aligned}
$$

Therefore,

$$
\Pr(\alpha<0) \leq 1 - \frac{\mathbb{E}_{S\sim\mathcal{D}\otimes 2k}\mathbb{E}_{c_1,c_2\sim\tau}\mathbb{E}_{\boldsymbol{x}\sim\mathcal{D}_c}[\alpha]^2}{\mathbb{E}_{c_1,c_2}[\mathrm{Var}_{\boldsymbol{x}\sim D_{c_1},S}[\alpha]+\mathbb{E}_{\boldsymbol{x}\sim D_{c_1},S}[\alpha]^2]}.
\tag{15}
$$

We write down the expection and variance of $\alpha$ with following Lemmas 3 and 4.

**Lemma 3.** *Under the same notation and assumptions as Theorem 2, then,*

$$
\mathbb{E}_{S\sim\mathcal{D}\otimes 2k}\mathbb{E}_{\boldsymbol{x}\sim\mathcal{D}_{c_1}}[\alpha] = (\boldsymbol{\mu}_{c_1}-\boldsymbol{\mu}_{c_2})^\top(\boldsymbol{\mu}_{c_1}-\boldsymbol{\mu}_{c_2})
$$
$$
\mathbb{E}_{c_1,c_2\sim\tau}\mathbb{E}_{\boldsymbol{x}\sim\mathcal{D}_{c_1}}\mathbb{E}_{S\sim\mathcal{D}\otimes 2K}[\alpha] = 2\mathrm{Tr}(\Sigma).
$$

**Lemma 4.** *Under the same notation and assumptions as Theorem 2, then,*

$$
\mathbb{E}_{c_1,c_2}\left[\mathrm{Var}_{\boldsymbol{x},S}\left[\alpha|c_1,c_2\right]\right] \leq 8\left(1+\frac{1}{K}\right)\mathrm{Tr}\left(\Sigma_\tau((1+\frac{1}{K})\Sigma_\tau+2\Sigma)\right).
\tag{16}
$$

The proofs of the above lemmas are in the current study [2].

With Proposition 1 and Lemma 3, Lemma 4, we obtain Theorem 2.

### A.4 Derivation Details of Theorem 1

We will describe the detailed derivation of Theorem 1 in this section. Our derivation is different from Cao et al. [2]'s study in the following points.

1. We re-derived Lemma 3 because the term of the difference between the trace of the class covariance matrices is erased in the lemma. This term cannot omit in our derivation since we do not assume the class covariance matrix to be the same among classes.

2. We re-derived the bound on the variance of squared Euclidean distance of two vectors, e.g Lemma 4. The derivation of Cao et al. [2] uses the property of quadratic forms of normally distributed random variables and the fact that the sum of normally distributed random variables is also distributed in Gaussian distribution. The calculation of the variance of squared $L_2$-norm without depending on the property of some distributions is not straightforward [38]. We divide the variance of squared Euclidean distance of two vectors into the variance of the norm of the feature vectors and the variance of the inner-product of vectors. Then we apply Cauchy–Schwarz inequality to the inner-product.

We start the proof from equation 15. We first prove the following Lemma 5 related to the expectation statistics of $\alpha$ in equation 15.

**Lemma 5.** *Under the same notations and assumptions as Theorem 1, then,*

$$\mathbb{E}_{S \sim \mathcal{D}^{\otimes 2k}} \mathbb{E}_{\boldsymbol{x} \sim \mathcal{D}_{c_1}} [\alpha] = \frac{1}{K} \left( \text{Tr}(\Sigma_{c_2}) - \text{Tr}(\Sigma_{c_1}) \right) + (\boldsymbol{\mu}_{c_1} - \boldsymbol{\mu}_{c_2})\top(\boldsymbol{\mu}_{c_1} - \boldsymbol{\mu}_{c_2})$$

$$\mathbb{E}_{c_1,c_2 \sim \tau} \mathbb{E}_{\boldsymbol{x} \sim \mathcal{D}_{c_1}} \mathbb{E}_{S \sim \mathcal{D}^{\otimes 2K}} [\alpha] = 2\text{Tr}(\Sigma).$$

*Proof.* First, from the definition of $\alpha$, we split $\mathbb{E}_{S \sim \mathcal{D}^{\otimes 2k}} \mathbb{E}_{\boldsymbol{x} \sim \mathcal{D}_{c_1}} [\alpha]$ in to two parts and examine them seperately.

$$\mathbb{E}_{S \sim \mathcal{D}^{\otimes 2k}} \mathbb{E}_{\boldsymbol{x} \sim \mathcal{D}_{c_1}} [\alpha] = \underbrace{\mathbb{E} \left[ \left\| \phi(\boldsymbol{x}) - \overline{\phi(S_{c_2})} \right\|^2 \right]}_{(i)} - \underbrace{\mathbb{E} \left[ \left\| \phi(\boldsymbol{x}) - \overline{\phi(S_{c_1})} \right\|^2 \right]}_{(ii)}. \tag{17}$$

In regular conditions, for random vector $X$, the expectation of the norm is

$$\mathbb{E}[X^\top X] = \text{Tr}(\text{Var}(X)) + \mathbb{E}[X]^\top \mathbb{E}[X], \tag{18}$$

and the variance of the vector is

$$\text{Var}(X) = \mathbb{E}[XX^\top] - \mathbb{E}[X]E[X]^\top \tag{19}$$

$$\Sigma_{c_i} \overset{\Delta}{=} \text{Var}_{\boldsymbol{x} \sim \mathcal{D}_{c_i}}(\phi(\boldsymbol{x})). \tag{20}$$

Hence,

$$(i) = \mathbb{E}_{\boldsymbol{x} \sim D_{c_1}} \mathbb{E}_S \left[ \left\| \phi(\boldsymbol{x}) - \overline{\phi(S_{c_2})} \right\|^2 \right]$$

$$= \text{Tr} \left( \text{Var}_{\boldsymbol{x} \sim D_{c_1}, S} \left[ \phi(\boldsymbol{x}) - \overline{\phi(S_{c_2})} \right] \right) + \mathbb{E}_{\boldsymbol{x}} \mathbb{E}_S \left[ \phi(\boldsymbol{x}) - \overline{\phi(S_{c_2})} \right]^\top \mathbb{E}_{\boldsymbol{x}} \mathbb{E}_S \left[ \phi(\boldsymbol{x}) - \overline{\phi(S_{c_2})} \right], \tag{21}$$

where the first term inside the trace can be expanded as:

$$\text{Var} \left[ \phi(\boldsymbol{x}) - \overline{\phi(S_{c_2})} \right] = \mathbb{E} \left[ \left( \phi(\boldsymbol{x}) - \overline{\phi(S_{c_2})} \right) \left( \phi(\boldsymbol{x}) - \overline{\phi(S_{c_2})} \right)^\top \right] - (\boldsymbol{\mu}_{c_1} - \boldsymbol{\mu}_{c_2})(\boldsymbol{\mu}_{c_1} - \boldsymbol{\mu}_{c_2})^\top$$

$$= \text{Var} \left[ \phi(\boldsymbol{x}) \right] + \mathbb{E} \left[ \phi(\boldsymbol{x}) \right] \mathbb{E} \left[ \phi(\boldsymbol{x}) \right]^\top + \text{Var} \left[ \overline{\phi(S_{c_2})} \right] + \mathbb{E} \left[ \overline{\phi(S_{c_2})} \right] \mathbb{E} \left[ \overline{\phi(S_{c_2})} \right]^\top$$

$$- \boldsymbol{\mu}_{c_2} \boldsymbol{\mu}_{c_1}^\top - \boldsymbol{\mu}_{c_1} \boldsymbol{\mu}_{c_2}^\top - (\boldsymbol{\mu}_{c_1} - \boldsymbol{\mu}_{c_2})(\boldsymbol{\mu}_{c_1} - \boldsymbol{\mu}_{c_2})^\top$$

$$= \Sigma_{c_1} + \frac{1}{K} \Sigma_{c_2} \quad \text{(Last terms cancel out).} \tag{22}$$

The second term in equation 21 is simply as follows.

$$\mathbb{E}_{\boldsymbol{x} \sim D_{c_1}} \mathbb{E}_S \left[ \phi(\boldsymbol{x}) - \overline{\phi(S_{c_2})} \right] = \boldsymbol{\mu}_{c_1} - \boldsymbol{\mu}_{c_2}. \tag{23}$$

From equation 22 and equation 23 we obtain

$$(i) = \text{Tr}(\Sigma_{c_1}) + \frac{1}{K} \text{Tr}(\Sigma_{c_2}) + (\boldsymbol{\mu}_{c_1} - \boldsymbol{\mu}_{c_2})^\top (\boldsymbol{\mu}_{c_1} - \boldsymbol{\mu}_{c_2}). \tag{24}$$

Similarly for ii,

$$(ii) = \text{Tr} \left( \text{Var}_{\boldsymbol{x} \sim D_{c_1}, S} \left[ \phi(\boldsymbol{x}) - \overline{\phi(S_{c_1})} \right] \right) + \mathbb{E}_{\boldsymbol{x}} \mathbb{E}_S \left[ \phi(\boldsymbol{x}) - \overline{\phi(S_{c_1})} \right]^\top \mathbb{E}_{\boldsymbol{x}} \mathbb{E}_S [\phi(\boldsymbol{x}) - \overline{\phi(S_{c_1})}]$$

$$= \text{Tr}(\Sigma_{c_1}) + \frac{1}{K} \text{Tr}(\Sigma_{c_1}). \tag{25}$$

From $(i)$ and $(ii)$, and equation 17

$$\mathbb{E}_{S \sim \mathcal{D}^{\otimes 2k}} \mathbb{E}_{\boldsymbol{x} \sim \mathcal{D}_{c_1}} [\alpha] = \frac{1}{K} \left( \text{Tr} (\Sigma_{c_2}) - \text{Tr} (\Sigma_{c_1}) \right) + \left( \boldsymbol{\mu}_{c_1} - \boldsymbol{\mu}_{c_2} \right)^\top \left( \boldsymbol{\mu}_{c_1} - \boldsymbol{\mu}_{c_2} \right). \tag{26}$$

Since $\mathbb{E}_{c_1,c_2,\boldsymbol{x},S}[\alpha] = \mathbb{E}_{c_1,c_2\sim\tau}[\mathbb{E}_{S\sim\mathcal{D}^{\otimes 2k}}\mathbb{E}_{\boldsymbol{x}\sim\mathcal{D}_{c_1}}[\alpha]]$,

$$
\begin{aligned}
\mathbb{E}_{c_1,c_2,\boldsymbol{x},S}[\alpha] &= \mathbb{E}_{c_1,c_2\sim\tau}\left[\frac{1}{K}(\mathrm{Tr}(\Sigma_{c_2}) - \mathrm{Tr}(\Sigma_{c_1})) + \left(\boldsymbol{\mu}_{c_1} - \boldsymbol{\mu}_{c_2}\right)^{\top}\left(\boldsymbol{\mu}_{c_1} - \boldsymbol{\mu}_{c_2}\right)\right] \\
&= \mathbb{E}_{c_1,c_2\sim\tau}\left[\left(\boldsymbol{\mu}_{c_1} - \boldsymbol{\mu}_{c_2}\right)^{\top}\left(\boldsymbol{\mu}_{c_1} - \boldsymbol{\mu}_{c_2}\right)\right] \\
&= \mathbb{E}_{c_1,c_2\sim\tau}\left[\boldsymbol{\mu}_{c_1}^{\top}\boldsymbol{\mu}_{c_1} + \boldsymbol{\mu}_{c_2}^{\top}\boldsymbol{\mu}_{c_2} - \boldsymbol{\mu}_{c_1}^{\top}\boldsymbol{\mu}_{c_2} - \boldsymbol{\mu}_{c_2}^{\top}\boldsymbol{\mu}_{c_1}\right] \\
&= \mathrm{Tr}\left(\Sigma\right) + \boldsymbol{\mu}^{\top}\boldsymbol{\mu} + \mathrm{Tr}\left(\Sigma\right) + \boldsymbol{\mu}^{\top}\boldsymbol{\mu} - 2\boldsymbol{\mu}^{\top}\boldsymbol{\mu} \\
&= 2\mathrm{Tr}\left(\Sigma\right).
\end{aligned}
\tag{27}
$$

$\square$

Next we prove the following Lemma 6 related to the conditioned variance of $\alpha$.

**Lemma 6.** *Under the same notation and assumptions as Theorem 1,*

$$
\mathbb{E}_{c_1,c_2}\mathrm{Var}_{\boldsymbol{x}\sim D_{c_1},S\sim\mathcal{D}^{\otimes 2N}}[\alpha] \leq \frac{4}{K}\mathbb{E}_{c\sim\tau}\mathrm{Var}\left[\|\phi(\boldsymbol{x})\|^2\right] + \frac{4}{K}\mathrm{Var}_{c\sim\tau}\left[\mathrm{Tr}(\Sigma_c)\right] + \mathrm{V_{wit}}(\Sigma_\tau, \Sigma, \boldsymbol{\mu}),
\tag{28}
$$

*where*

$$
\begin{aligned}
\mathrm{V_{wit}}\left(\Sigma_\tau, \Sigma, \boldsymbol{\mu}\right) &= \frac{8}{K}\left(\mathrm{Tr}(\Sigma_\tau)\right)\left(\mathrm{Tr}(\Sigma) + \boldsymbol{\mu}^{\top}\boldsymbol{\mu}\right) + 4\left(\mathrm{Tr}(\Sigma) + \boldsymbol{\mu}^{\top}\boldsymbol{\mu}\right)^2 \\
&\quad + 4\mathbb{E}_{c_1,c_2\sim\tau}\left[\mathrm{Tr}\left(\Sigma_{c_1}\right)\left(\boldsymbol{\mu}_{c_2} - \boldsymbol{\mu}_{c_1}\right)^{\top}\left(\boldsymbol{\mu}_{c_2} - \boldsymbol{\mu}_{c_1}\right)\right].
\end{aligned}
\tag{29}
$$

*Proof.* We start with the inequality between the variance of 2 random variables. We define $\mathrm{Cov}(A, B)$ as covariance of 2 random variables $A, B$.

$$
\begin{aligned}
\mathrm{Var}[A + B] &= \mathrm{Var}[A] + \mathrm{Var}[B] + 2\mathrm{Cov}(A, B) \\
&\leq \mathrm{Var}[A] + \mathrm{Var}[B] + 2\sqrt{\mathrm{Var}[A]\mathrm{Var}[B]} \\
&\leq \mathrm{Var}[A] + \mathrm{Var}[B] + 2\cdot\frac{\mathrm{Var}[A] + \mathrm{Var}[B]}{2} \\
&= 2\mathrm{Var}[A] + 2\mathrm{Var}[B].
\end{aligned}
\tag{30}
$$

For $\mathrm{Var}_{\boldsymbol{x}\sim D_{c_1},S}[\alpha]$,

$$
\begin{aligned}
\mathrm{Var}_{\boldsymbol{x}\sim D_{c_1},S}[\alpha] &= \mathrm{Var}\left[\left\|\phi(\boldsymbol{x}) - \overline{\phi(S_{c_2})}\right\|^2 - \left\|\phi(\boldsymbol{x}) - \overline{\phi(S_{c_1})}\right\|^2\right] \\
&= \mathrm{Var}\left[\left\|\overline{\phi(S_{c_1})}\right\|^2 - \left\|\overline{\phi(S_{c_2})}\right\|^2 - 2\phi(\boldsymbol{x})^{\top}\left(\overline{\phi(S_{c_2})} - \overline{\phi(S_{c_1})}\right)\right] \\
&\leq 2\mathrm{Var}\left[\left\|\overline{\phi(S_{c_1})}\right\|^2 - \left\|\overline{\phi(S_{c_2})}\right\|^2\right] \\
&\quad + 4\mathrm{Var}\left[\phi(\boldsymbol{x})^{\top}\left(\overline{\phi(S_{c_2})} - \overline{\phi(S_{c_1})}\right)\right] \quad (\because equation\ 30) \\
&= 2\mathrm{Var}\left[\left\|\overline{\phi(S_{c_1})}\right\|^2\right] + 2\mathrm{Var}\left[\left\|\overline{\phi(S_{c_2})}\right\|^2\right] + 4\mathrm{Var}\left[\phi(\boldsymbol{x})^{\top}(\overline{\phi(S_{c_2})} - \overline{\phi(S_{c_1})})\right].
\end{aligned}
\tag{31}
$$

From 3rd line to 4th line we decompose the variance of $\left\|\overline{\phi(S_{c_1})}\right\|^2 - \left\|\overline{\phi(S_{c_2})}\right\|^2$ use the independence of $\overline{\phi(S_{c_1})}$ and $\overline{\phi(S_{c_2})}$ with their class given.

Next we focus on $\mathrm{Var}\left[\phi(\boldsymbol{x})^{\top}(\overline{\phi(S_{c_2})} - \overline{\phi(S_{c_1})})\right]$.

$$
\begin{aligned}
\mathrm{Var}\left[\phi(\boldsymbol{x})^{\top}(\overline{\phi(S_{c_2})} - \overline{\phi(S_{c_1})})\right] &= \mathbb{E}\left[\left(\phi(\boldsymbol{x})^{\top}\left(\overline{\phi(S_{c_2})} - \overline{\phi(S_{c_1})}\right)\right)^2\right] \\
&\quad - \left(\mathbb{E}\left[\phi(\boldsymbol{x})\right]^{\top}\mathbb{E}\left[\left(\overline{\phi(S_{c_2})} - \overline{\phi(S_{c_1})}\right)\right]\right)^2 \\
&\leq \mathbb{E}\left[\left(\|\phi(\boldsymbol{x})\|^2\left\|\overline{\phi(S_{c_2})} - \overline{\phi(S_{c_1})}\right\|\right)^2\right] - (\boldsymbol{\mu}_{c_1}^{\top}(\boldsymbol{\mu}_{c_2} - \boldsymbol{\mu}_{c_1}))^2 \\
&= \mathbb{E}\left[\|\phi(\boldsymbol{x})\|^2\right]\mathbb{E}\left[\left\|\overline{\phi(S_{c_2})} - \overline{\phi(S_{c_1})})\right\|^2\right] - (\boldsymbol{\mu}_{c_1}^{\top}(\boldsymbol{\mu}_{c_2} - \boldsymbol{\mu}_{c_1}))^2.
\end{aligned}
$$
$$(32)$$

From the 2nd line to the 3rd line we use Cauchy–Schwarz inequality.

For $\mathbb{E}\left[\left\|\overline{\phi(S_{c_2})} - \overline{\phi(S_{c_1})}\right\|^2\right]$, with equation 18

$$
\mathbb{E}\left[\left\|\overline{\phi(S_{c_2})} - \overline{\phi(S_{c_1})}\right\|^2\right] = \frac{1}{K}\left(\mathrm{Tr}\left(\Sigma_{c_1}\right) + \mathrm{Tr}\left(\Sigma_{c_2}\right)\right) + \left(\boldsymbol{\mu}_{c_2} - \boldsymbol{\mu}_{c_1}\right)^{\top}\left(\boldsymbol{\mu}_{c_2} - \boldsymbol{\mu}_{c_1}\right). \tag{33}
$$

Thus $\mathbb{E}_{c_1,c_2\sim\tau}\left[\mathbb{E}_{\boldsymbol{x}\sim\mathcal{D}_{c_1},S}\left[\|\phi(\boldsymbol{x})\|^2\right]\mathbb{E}_{\boldsymbol{x}\sim\mathcal{D}_{c_1},S}\left[\left\|\overline{\phi(S_{c_2})} - \overline{\phi(S_{c_1})}\right\|^2\right]\right]$ is calculated as follows.

$$
\begin{aligned}
&\mathbb{E}_{c_1,c_2\sim\tau}\left[\mathbb{E}_{\boldsymbol{x}\sim\mathcal{D}_{c_1},S}\left[\|\phi(\boldsymbol{x})\|^2\right]\mathbb{E}_{\boldsymbol{x}\sim\mathcal{D}_{c_1},S}\left[\left\|\overline{\phi(S_{c_2})} - \overline{\phi(S_{c_1})}\right\|^2\right]\right] \\
&= \mathbb{E}_{c_1,c_2\sim\tau}\left[\left(\mathrm{Tr}(\Sigma_{c_1}) + \boldsymbol{\mu}_{c_1}^{\top}\boldsymbol{\mu}_{c_1}\right)\left(\frac{1}{K}\left(\mathrm{Tr}(\Sigma_{c_1}) + \mathrm{Tr}(\Sigma_{c_2})\right) + (\boldsymbol{\mu}_{c_2} - \boldsymbol{\mu}_{c_1})^{\top}(\boldsymbol{\mu}_{c_2} - \boldsymbol{\mu}_{c_1})\right)\right] \\
&= \mathbb{E}_{c_1,c_2\sim\tau}\left[\frac{1}{K}\left(\mathrm{Tr}\left(\Sigma_{c_1}\right)^2 + \mathrm{Tr}(\Sigma_{c_1})\mathrm{Tr}\left(\Sigma_{c_2}\right)\right)\right] \\
&\quad + \mathbb{E}_{c_1,c_2\sim\tau}\left[\frac{2}{K}\left(\mathrm{Tr}\left(\Sigma_{\tau}\right)\right)\boldsymbol{\mu}_{c_1}^{\top}\boldsymbol{\mu}_{c_1} + \boldsymbol{\mu}_{c_1}^{\top}\boldsymbol{\mu}_{c_1}\left(\boldsymbol{\mu}_{c_2} - \boldsymbol{\mu}_{c_1}\right)^{\top}\left(\boldsymbol{\mu}_{c_2} - \boldsymbol{\mu}_{c_1}\right)\right] \\
&\quad + \mathbb{E}_{c_1,c_2\sim\tau}\left[\mathrm{Tr}\left(\Sigma_{c_1}\right)\left(\boldsymbol{\mu}_{c_2} - \boldsymbol{\mu}_{c_1}\right)^{\top}\left(\boldsymbol{\mu}_{c_2} - \boldsymbol{\mu}_{c_1}\right)\right] \\
&= \frac{1}{K}\left(\mathbb{E}_{c_1,c_2\sim\tau}\left[\mathrm{Tr}(\Sigma_{c_1})^2\right] + \mathbb{E}_{c_1,c_2\sim\tau}\left[\mathrm{Tr}(\Sigma_{c_1})\right]^2\right) \\
&\quad + \frac{2}{K}\left(\mathrm{Tr}\left(\Sigma_{\tau}\right)\right)\left(\mathrm{Tr}\left(\Sigma\right) + \boldsymbol{\mu}^{\top}\boldsymbol{\mu}\right) + \mathbb{E}\left[\boldsymbol{\mu}_{c_1}^{\top}\boldsymbol{\mu}_{c_1}\left(\boldsymbol{\mu}_{c_2} - \boldsymbol{\mu}_{c_1}\right)^{\top}\left(\boldsymbol{\mu}_{c_2} - \boldsymbol{\mu}_{c_1}\right)\right] \\
&\quad + \mathbb{E}_{c_1,c_2\sim\tau}\left[\mathrm{Tr}\left(\Sigma_{c_1}\right)\left(\boldsymbol{\mu}_{c_2} - \boldsymbol{\mu}_{c_1}\right)^{\top}\left(\boldsymbol{\mu}_{c_2} - \boldsymbol{\mu}_{c_1}\right)\right] \\
&= \frac{1}{K}\mathrm{Var}_{c\sim\tau}\left[\mathrm{Tr}\left(\Sigma_c\right)\right] \\
&\quad + \frac{2}{K}\mathrm{Tr}\left(\Sigma_{\tau}\right)^2 + \frac{2}{K}\left(\mathrm{Tr}\left(\Sigma_{\tau}\right)\right)\left(\mathrm{Tr}\left(\Sigma\right) + \boldsymbol{\mu}^{\top}\boldsymbol{\mu}\right) + \mathbb{E}_{c_1,c_2}\left[\boldsymbol{\mu}_{c_1}^{\top}\boldsymbol{\mu}_{c_1}\left(\boldsymbol{\mu}_{c_2} - \boldsymbol{\mu}_{c_1}\right)^{\top}\left(\boldsymbol{\mu}_{c_2} - \boldsymbol{\mu}_{c_1}\right)\right] \\
&\quad + \mathbb{E}_{c_1,c_2\sim\tau}\left[\mathrm{Tr}\left(\Sigma_{c_1}\right)\left(\boldsymbol{\mu}_{c_2} - \boldsymbol{\mu}_{c_1}\right)^{\top}\left(\boldsymbol{\mu}_{c_2} - \boldsymbol{\mu}_{c_1}\right)\right] \\
&= \frac{1}{K}\mathrm{Var}_{c\sim\tau}\left[\mathrm{Tr}(\Sigma_c)\right] \\
&\quad + \frac{2}{K}\mathrm{Tr}(\Sigma_{\tau})\left(\mathrm{Tr}(\Sigma_{\tau}) + \mathrm{Tr}(\Sigma) + \boldsymbol{\mu}^{\top}\boldsymbol{\mu}\right) + \mathbb{E}_{c_1,c_2}\left[\boldsymbol{\mu}_{c_1}^{\top}\boldsymbol{\mu}_{c_1}\left(\boldsymbol{\mu}_{c_2} - \boldsymbol{\mu}_{c_1}\right)^{\top}\left(\boldsymbol{\mu}_{c_2} - \boldsymbol{\mu}_{c_1}\right)\right] \\
&\quad + \mathbb{E}_{c_1,c_2\sim\tau}\left[\mathrm{Tr}\left(\Sigma_{c_1}\right)\left(\boldsymbol{\mu}_{c_2} - \boldsymbol{\mu}_{c_1}\right)^{\top}\left(\boldsymbol{\mu}_{c_2} - \boldsymbol{\mu}_{c_1}\right)\right].
\end{aligned}
$$
$$(34)$$

Now we take into account the term $-(\boldsymbol{\mu}_{c_1}^\top(\boldsymbol{\mu}_{c_2} - \boldsymbol{\mu}_{c_1}))^2$ in equation 32,

$$
\begin{aligned}
\mathbb{E}_{c_1,c_2} & \left[\boldsymbol{\mu}_{c_1}^\top \boldsymbol{\mu}_{c_1}(\boldsymbol{\mu}_{c_2} - \boldsymbol{\mu}_{c_1})^\top(\boldsymbol{\mu}_{c_2} - \boldsymbol{\mu}_{c_1}) - (\boldsymbol{\mu}_{c_1}^\top(\boldsymbol{\mu}_{c_2} - \boldsymbol{\mu}_{c_1}))^2\right] \\
&= \mathbb{E}_{c_1,c_2}\left[\boldsymbol{\mu}_{c_1}^\top \boldsymbol{\mu}_{c_1}\boldsymbol{\mu}_{c_2}^\top \boldsymbol{\mu}_{c_2} - (\boldsymbol{\mu}_{c_1}^\top\boldsymbol{\mu}_{c_2})^2\right] \\
&\leq \mathbb{E}_{c_1,c_2}\left[\boldsymbol{\mu}_{c_1}^\top \boldsymbol{\mu}_{c_1}\boldsymbol{\mu}_{c_2}^\top \boldsymbol{\mu}_{c_2}\right] \\
&= \left(\mathrm{Tr}(\Sigma) + \boldsymbol{\mu}^\top\boldsymbol{\mu}\right)^2 .
\end{aligned}
\tag{35}
$$

Thus $\mathbb{E}_{c_1,c_2}\left[\mathrm{Var}[\phi(\boldsymbol{x})^\top(\overline{\phi(S_{c_2})} - \overline{\phi(S_{c_1})})]\right]$ is calculated as follows.

$$
\begin{aligned}
& \mathbb{E}_{c_1,c_2}\left[\mathrm{Var}[\phi(\boldsymbol{x})^\top(\overline{\phi(S_{c_2})} - \overline{\phi(S_{c_1})})]\right] \\
&= \frac{1}{K}\mathrm{Var}_{c\sim\tau}\left[\mathrm{Tr}(\Sigma_c)\right] + \frac{2}{K}\mathrm{Tr}(\Sigma_\tau)\left(\mathrm{Tr}(\Sigma_\tau) + \mathrm{Tr}(\Sigma) + \boldsymbol{\mu}^\top\boldsymbol{\mu}\right) + (\mathrm{Tr}(\Sigma) + \boldsymbol{\mu}^\top\boldsymbol{\mu})^2 \\
&\quad + \mathbb{E}_{c_1,c_2\sim\tau}\left[\mathrm{Tr}(\Sigma_{c_1})(\boldsymbol{\mu}_{c_2} - \boldsymbol{\mu}_{c_1})^\top(\boldsymbol{\mu}_{c_2} - \boldsymbol{\mu}_{c_1})\right] .
\end{aligned}
\tag{36}
$$

Regarding $\left\|\overline{\phi(S_c)}\right\|^2$, since the function computing square norm is convex, next equation holds with $D$-dimensional Jensen's inequality [39]:

$$
\left\|\overline{\phi(S_c)}\right\|^2 = \left\|\frac{1}{K}\sum_{\substack{i=0 \\ x\in S_c}}\phi(\boldsymbol{x})\right\|^2
$$

$$
\leq \frac{1}{K}\left\|\sum_{\substack{i=0 \\ x\in S_c}}\phi(\boldsymbol{x})\right\|^2 .
\tag{37}
$$

Combining equation 31, equation 36, and equation 37 we obtain

$$
\begin{aligned}
& \mathbb{E}_{c_1,c_2}\mathrm{Var}_{\boldsymbol{x}\sim D_{c_1}, S\sim\mathcal{D}^{\otimes 2N}}[\alpha] \\
&\leq \frac{4}{K}\mathbb{E}_{c\sim\tau}\mathrm{Var}_{\boldsymbol{x}\sim\mathcal{D}_c}\left[\|\phi(\boldsymbol{x})\|^2\right] + \frac{4}{K}\mathrm{Var}_{c\sim\tau}\left[\mathrm{Tr}(\Sigma_c)\right] \\
&\quad + \frac{8}{K}\mathrm{Tr}(\Sigma_\tau)\left(\mathrm{Tr}(\Sigma_\tau) + \mathrm{Tr}(\Sigma) + \boldsymbol{\mu}^\top\boldsymbol{\mu}\right) + 4\left(\mathrm{Tr}(\Sigma) + \boldsymbol{\mu}^\top\boldsymbol{\mu}\right)^2 \\
&\quad + 4\mathbb{E}_{c_1,c_2\sim\tau}\left[\mathrm{Tr}(\Sigma_{c_1})(\boldsymbol{\mu}_{c_2} - \boldsymbol{\mu}_{c_1})^\top(\boldsymbol{\mu}_{c_2} - \boldsymbol{\mu}_{c_1})\right] .
\end{aligned}
\tag{38}
$$

$\square$

To complete the proof of Theorem 1, we calculate $\mathbb{E}_{c_1,c_2}\mathbb{E}_{\boldsymbol{x}\sim\mathcal{D}_{c_1}, S\sim\mathcal{D}^{\otimes 2K}}[\alpha]^2$.

$$
\begin{aligned}
& \mathbb{E}_{c_1,c_2}\mathbb{E}_{\boldsymbol{x}\sim\mathcal{D}_{c_1}, S\sim\mathcal{D}^{\otimes 2K}}[\alpha]^2 \\
&= \mathbb{E}_{c_1,c_2}\left[\left(\frac{1}{K}(\mathrm{Tr}(\Sigma_{c_2}) - \mathrm{Tr}(\Sigma_{c_1})) + (\boldsymbol{\mu}_{c_1} - \boldsymbol{\mu}_{c_2})^\top(\boldsymbol{\mu}_{c_1} - \boldsymbol{\mu}_{c_2})\right)^2\right] \\
&= \mathbb{E}_{c_1,c_2}\left[\left(\frac{1}{K}(\mathrm{Tr}(\Sigma_{c_2}) - \mathrm{Tr}(\Sigma_{c_1}))\right)^2\right] + \mathbb{E}_{c_1,c_2}\left[\left((\boldsymbol{\mu}_{c_1} - \boldsymbol{\mu}_{c_2})^\top(\boldsymbol{\mu}_{c_1} - \boldsymbol{\mu}_{c_2})\right)^2\right] \\
&\quad + 2\mathbb{E}_{c_1,c_2}\left[\left(\frac{1}{K}(\mathrm{Tr}(\Sigma_{c_2}) - \mathrm{Tr}(\Sigma_{c_1}))\right)\left((\boldsymbol{\mu}_{c_1} - \boldsymbol{\mu}_{c_2})^\top(\boldsymbol{\mu}_{c_1} - \boldsymbol{\mu}_{c_2})\right)\right] \\
&= \mathrm{Var}_{c_1,c_2\sim\tau}\left[\frac{1}{K}(\mathrm{Tr}(\Sigma_{c_2}) - \mathrm{Tr}(\Sigma_{c_1}))\right] + \left(\mathbb{E}_{c_1,c_2\sim\tau}\left[\frac{1}{K}(\mathrm{Tr}(\Sigma_{c_2}) - \mathrm{Tr}(\Sigma_{c_1}))\right]\right)^2 \\
&\quad + \mathbb{E}_{c_1,c_2}\left[\left((\boldsymbol{\mu}_{c_1} - \boldsymbol{\mu}_{c_2})^\top(\boldsymbol{\mu}_{c_1} - \boldsymbol{\mu}_{c_2})\right)^2\right] \\
&= \frac{2}{K^2}\mathrm{Var}_{c\sim\tau}\left[\mathrm{Tr}(\Sigma_c)\right] + \mathbb{E}_{c_1,c_2}\left[\left((\boldsymbol{\mu}_{c_1} - \boldsymbol{\mu}_{c_2})^\top(\boldsymbol{\mu}_{c_1} - \boldsymbol{\mu}_{c_2})\right)^2\right] .
\end{aligned}
\tag{39}
$$

From 2nd line to 3rd line, we use the symmetry of the last term with respect to $c_1$ and $c_2$ and erase the term.

Combining equation 15, Lemma 5, Lemma 6, and equation 39, we obtain the bound.

## A.5 Theorem 1 with *N*-way Classification

The upper bound on the risk of $N$-way prototype classifier is as follows.

**Theorem 7.** *Let operation of binary class prototype classifier $\mathcal{M}$ as defined in equation 1. Then for $\overline{\phi(S_c)} = \frac{1}{K}\Sigma_{\boldsymbol{x}\in S_c}\phi(\boldsymbol{x})$, $\mu_c = \mathbb{E}_{\boldsymbol{x}\sim\mathcal{D}_c}[\phi(\boldsymbol{x})]$, $\Sigma_c = \mathbb{E}_{\boldsymbol{x}\sim\mathcal{D}_c}[(\phi(\boldsymbol{x})-\mu_c)(\phi(\boldsymbol{x})-\mu_c)^\top]$, $\mu = \mathbb{E}_{c\sim\tau}[\mu_c]$, $\Sigma = \mathbb{E}_{c\sim\tau}[(\mu_c-\mu)(\mu_c-\mu)^\top]$, $\Sigma_\tau = \mathbb{E}_{c\sim\tau}[\Sigma_c]$, if $\phi(\boldsymbol{x})$ has its fourth moment, miss classification risk of binary class prototype classifier $\mathrm{R}_\mathcal{M}$ satisfy*

$$\mathrm{R}(\mathcal{M}(\phi, \boldsymbol{x}, \{S_i\}_{i=1}^N), y)$$

$$\leq N - 1 - \sum_{\substack{c=1 \\ c\neq y}}^{N} \frac{4(\mathrm{Tr}(\Sigma))^2}{\mathbb{E}\mathrm{V}[h_{L2}(\phi(\boldsymbol{x}))] + \mathrm{V}_{\mathrm{Tr}}(\Sigma_y) + \mathrm{V}_{\mathrm{wit}}(\Sigma_\tau, \Sigma, \boldsymbol{\mu}) + \mathbb{E}\,\mathrm{dist}_{\mathrm{L2}}^2(\boldsymbol{\mu}_y, \boldsymbol{\mu}_c)}, \quad (40)$$

*where*

$$\mathbb{E}\mathrm{V}[h_{L2}(\phi(\boldsymbol{x}))] = \frac{4}{K}\mathbb{E}_{y\sim\tau}\left[\mathrm{Var}_{\boldsymbol{x}_c\sim\mathcal{D}_c}\left[\|\phi(\boldsymbol{x})\|^2\right]\right],$$

$$\mathrm{V}_{\mathrm{Tr}}(\Sigma_y) = \left(\frac{4}{K} + \frac{2}{K^2}\right)\mathrm{Var}_{c\sim\tau}\left[\mathrm{Tr}\left(\Sigma_c\right)\right],$$

$$\mathrm{V}_{\mathrm{wit}}\left(\Sigma_\tau, \Sigma, \boldsymbol{\mu}\right) = \frac{8}{K}\left(\mathrm{Tr}(\Sigma_\tau)\right)\left(\mathrm{Tr}(\Sigma) + \boldsymbol{\mu}^\top\boldsymbol{\mu}\right) + 4\left(\mathrm{Tr}(\Sigma) + \boldsymbol{\mu}^\top\boldsymbol{\mu}\right)^2$$

$$+ 4\mathbb{E}_{c\sim\tau}\left[\mathrm{Tr}\left(\Sigma_y\right)\left(\boldsymbol{\mu}_y - \boldsymbol{\mu}_c\right)^\top\left(\boldsymbol{\mu}_y - \boldsymbol{\mu}_c\right)\right], \quad (41)$$

$$\mathbb{E}\,\mathrm{dist}_{\mathrm{L2}}^2(\boldsymbol{\mu}_y, \boldsymbol{\mu}_c) = \mathbb{E}_{y,c}\left[\left(\left(\boldsymbol{\mu}_y - \boldsymbol{\mu}_c\right)^\top\left(\boldsymbol{\mu}_y - \boldsymbol{\mu}_c\right)\right)^2\right].$$

*Proof.* Let $x, y$ be the input and its class of a query data. We define $\alpha_c$ by $\alpha_c = \left\|\phi(\boldsymbol{x}) - \overline{\phi(S_c)}\right\|^2 - \left\|\phi(\boldsymbol{x}) - \overline{\phi(S_y)}\right\|^2$. Then a prototype classifier miss-classify a class of input $x$, $\hat{y}$, if $\exists c \in [1, N], c \neq y, \alpha_c < 0$. Hence: $\mathrm{R}_\mathcal{M}(\phi) = \mathrm{Pr}(\bigcup_{\substack{c=1 \\ c\neq y}}^N \alpha_c < 0)$

By Frechet's inequality, next equation holds:

$$\mathrm{R}_\mathcal{M}(\phi) \leq \sum_{\substack{c=1 \\ c\neq y}}^{N} \mathrm{Pr}(\alpha_i < 0).$$

Noting that Theorem 1 can be applied to each term in the summation and then we obtain Theorem 7. $\qquad\square$

## A.6 Detailed performance results

We show in Table 5 the detailed performance results of standard object recognition in this section with 95% confidence margin. The table shows the similar result with Table 1. We can observe that the prototype classifier with $L_2$-norm, EST+$L_2$-norm, LDA+$L_2$-norm performs comparably with ProtoNet and the linear-evaluation-based approach in most of the settings.

Table 5: Classification accuracies with ResNet18 on *mini*ImageNet, *tiered*ImageNet, CIFARFS, and FC100 of ProtoNet, linear-evaluation-based methods [3], centering with $L_2$-norm [24], and ours. The Baseline without linear-evaluation methods with accuracy greater than the lower 95% confidence margin of the accuracy of ProtoNet and Baseline are in bold. Regarding to Baseline++, Baseline++ without linear-evaluation methods with accuracy greater than the lower 95% confidence margin of the accuracy of ProtoNet and Baseline++ are in bold.

| | *mini*ImageNet | | *tiered*ImageNet | | CIFAR-FS | | FC100 | |
|---|---|---|---|---|---|---|---|---|
| | 1-shot | 5-shot | 1-shot | 5-shot | 1-shot | 5-shot | 1-shot | 5-shot |
| *methods with meta-learning or linear-evaluation* | | | | | | | | |
| ProtoNet[1] | 56.74 ± 0.84 | 75.64 ± 0.62 | 61.75 ± 0.94 | 81.56 ± 0.68 | 65.46 ± 0.96 | 79.52 ± 0.66 | 35.92 ± 0.71 | 50.86 ± 0.71 |
| Baseline [3] | 55.41 ± 0.82 | 76.95 ± 0.61 | 63.38 ± 0.91 | 83.18 ± 0.64 | 65.12 ± 0.88 | 79.68 ± 0.64 | 40.06 ± 0.68 | 57.04 ± 0.71 |
| *A prototype classifier with feature-transformation methods* | | | | | | | | |
| Baseline-w/o-linear , centering+$L_2$-norm[24] | **57.67 ± 0.83** | **75.50 ± 0.67** | **65.26 ± 0.88** | **81.63 ± 0.64** | **65.26 ± 0.86** | **78.58 ± 0.66** | **41.51 ± 0.72** | **56.44 ± 0.74** |
| Baseline-w/o-linear | 43.86 ± 0.80 | 72.36 ± 0.92 | 56.16 ± 0.89 | 80.33 ± 0.66 | 54.06 ± 0.85 | 77.76 ± 0.64 | 35.90 ± 0.61 | 55.20 ± 0.77 |
| Baseline-w/o-linear , $L_2$-norm | **56.57 ± 0.80** | **76.44 ± 0.61** | **65.19 ± 0.87** | **82.93 ± 0.66** | **64.76 ± 0.87** | **79.90 ± 0.66** | **40.96 ± 0.71** | **57.71 ± 0.76** |
| Baseline-w/o-linear , EST | 44.19 ± 0.82 | 70.85 ± 0.92 | 57.05 ± 0.93 | 73.16 ± 0.78 | 57.16 ± 0.88 | 78.98 ± 0.69 | **48.43 ± 0.91** | **63.75 ± 0.73** |
| Baseline-w/o-linear , EST+$L_2$-norm | **56.39 ± 0.79** | **76.24 ± 0.64** | **64.71 ± 0.91** | **83.24 ± 0.67** | **65.54 ± 0.71** | **79.80 ± 0.71** | **50.50 ± 0.97** | **65.31 ± 0.77** |
| Baseline-w/o-linear , LDA | 47.14 ± 0.81 | 69.87 ± 0.65 | 57.05 ± 0.90 | 81.19 ± 0.65 | 56.49 ± 0.86 | 78.88 ± 0.66 | **50.29 ± 0.85** | **63.56 ± 0.77** |
| Baseline-w/o-linear , LDA+$L_2$-norm | **56.37 ± 0.81** | **76.39 ± 0.66** | **65.44 ± 0.94** | **83.16 ± 0.62** | **65.64 ± 0.84** | **80.72 ± 0.67** | **50.40 ± 0.97** | **65.31 ± 0.77** |
| *methods with meta-learning or linear-evaluation* | | | | | | | | |
| Baseline++ [3] | 55.07 ± 0.81 | 74.71 ± 0.61 | 64.02 ± 0.92 | 83.18 ± 0.64 | 65.64 ± 0.93 | 79.80 ± 0.66 | 36.93 ± 0.70 | 50.41 ± 0.73 |
| *A prototype classifier with feature-transformation methods* | | | | | | | | |
| Baseline++-w/o-linear , centering+$L_2$-norm[24] | **57.00 ± 0.64** | **74.06 ± 0.61** | **64.92 ± 0.91** | **81.41 ± 0.66** | **66.14 ± 0.93** | **80.03 ± 0.71** | **38.30 ± 0.74** | **51.06 ± 0.70** |
| Baseline++-w/o-linear | 36.80 ± 0.76 | 63.76 ± 0.71 | 48.27 ± 0.91 | 75.87 ± 0.71 | 50.18 ± 0.96 | 74.23 ± 0.70 | 30.76 ± 0.62 | 47.62 ± 0.69 |
| Baseline++-w/o-linear , $L_2$-norm | **57.21 ± 0.83** | **74.89 ± 0.65** | **66.67 ± 0.94** | **82.49 ± 0.68** | **66.84 ± 0.92** | **80.49 ± 0.71** | **38.55 ± 0.72** | **51.15 ± 0.71** |
| Baseline++-w/o-linear , EST | 47.21 ± 0.77 | 69.11 ± 0.64 | 53.49 ± 0.90 | 71.81 ± 0.75 | 53.36 ± 0.93 | 73.70 ± 0.74 | **39.92 ± 0.88** | **53.61 ± 0.70** |
| Baseline++-w/o-linear , EST+$L_2$-norm | **57.00 ± 0.75** | **76.13 ± 0.64** | **65.52 ± 0.97** | **79.96 ± 0.74** | **66.57 ± 0.95** | **79.42 ± 0.70** | **42.89 ± 0.82** | **56.82 ± 0.70** |
| Baseline++-w/o-linear , LDA | 45.95 ± 0.78 | 68.34 ± 0.68 | 49.09 ± 0.90 | 76.17 ± 0.68 | 52.98 ± 0.91 | 73.36 ± 0.75 | **40.27 ± 0.79** | **55.80 ± 0.74** |
| Baseline++-w/o-linear , LDA+$L_2$-norm | **56.78 ± 0.87** | **75.83 ± 0.65** | **66.68 ± 0.90** | **82.53 ± 0.67** | **65.89 ± 0.93** | **79.47 ± 0.70** | **44.21 ± 0.86** | **58.10 ± 0.70** |

## A.7 Visualization of the feature distribution

We show in Figure 3 the distribution of features on testset of FC100 before and after applying the data transformation. From figure 3, we can observe that the feature-transformation-methods slightly change the distributions even the transformations improve the performance of a prototype classifier. The projection into a lower dimensional space for visualization does not accurately represent the relationship of a higher dimension.

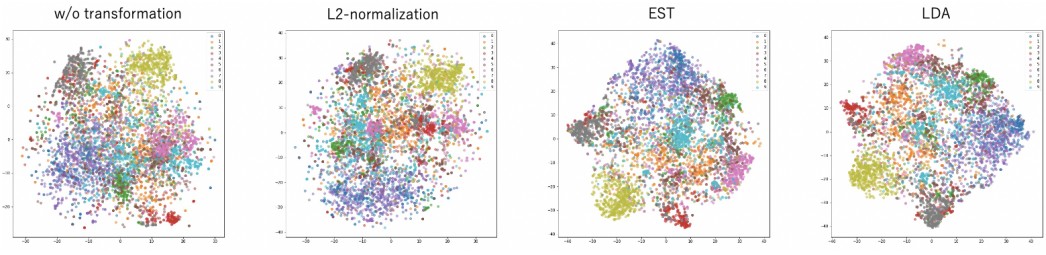

Figure 3