# OpenReview forum: "A Closer Look at Prototype Classifier for Few-shot Image Classification"
_NeurIPS.cc/2022/Conference — NeurIPS 2022 Accept_

### Official Review · Reviewer_bREF · 2022-06-20

**Rating:** 6
**Confidence:** 4
**Soundness:** 3 good
**Presentation:** 3 good
**Contribution:** 3 good

**Summary:**

The authors derive a more relaxed generalization bound for prototypical networks. Using the generalization bound, the authors propose the use of feature transformation to improve generalization of classifiers. The authors experiment with different normalization techniques on multiple few shot learning datasets and find that L2 based normalization can produce similar recognition performance as the one linear discriminant regularization.


**Questions:**

In Figure 1, don’t the shape of the clusters vary because Euclidean distance is used for ProtoNets while dot-product is used for Linear layers ? Is it because of the loss function ? The authors can elaborate on this further.

Can t-SNE feature visualization be shown to see how feature transformation affects discriminability ?

Can this feature transformation approach improve non-prototypical few shot learning methods ?

--------- POST REBUTTAL ----------

After rebuttal I have increased form borderline accept to weak accept.


**Limitations:**

No societal impacts


**Strengths And Weaknesses:**

— Strengths —

The authors derive a generalization bound that requires fewer assumptions about the data distribution family.

Experimental results on multiple datasets are consistently supported by their developed theorem.

The authors can produce reasonable amount of recognition performance without any fine-tuning and therefore can be useful for devices where backpropagation is not supported.

— Weaknesses —

The novelty of the proposed bound is incremental compared to the generalisation bound Cao et al. It might be good for the authors to elaborate point-wise differences between the bounds

Comparison studies have only been carried out against a few few shot learning methods.

More complex feature transformation methods should also have been explored such as exponential, non-linear kernels, power transforms etc to see whether they can also produce better generalization.

---

> ### Author Response · Authors · 2022-08-02
> **Response to Reviewer bREF part1**
>
> We would like to thank reviewer bREF for providing valuable feedback and raising interesting questions which we answer below.
>
> ### Q1. It might be good for the authors to elaborate point-wise differences between the bounds of the paper and the Cao’s.
> The conclusion and insights of our bound is also different from Cao’s bound as follows:
>
> - While Cao's work cannot explain why the L2-normalization of a feature vector can improve the performance of a prototype classifier, our work theoretically shows that the transformation can improve the performance.
> - We relax the assumption of Cao’s work; specifically, the bound does not require that the features be distributed in Gaussian distribution, and each covariance matrix does not have to be the same among classes. Therefore, our theoretical results can be applied to the features extracted by a model trained with cross-entropy loss with a linear classifier.
> - While Cao’s bound is depend only on the ratio of the within-class variance to the between-class variance, we theoretically show that the bound consists of three terms: (1) the variance of the norm of feature vectors, (2) the difference in the distribution shape constructed from each class embedding, and (3) the ratio of the within-class variance to the between-class variance. We experimentally show that reducing the term (1) and (3) respectively can improve the performance of a prototype classifier.
>
>
> ### C1. Comparison studies have only been carried out against a few shot learning methods.
> As we stated in Abstraction and Introduction, we focus on few-shot learning problems. Few-shot learning requires the model to quickly adapt to new classes. While the linear-evaluation method requires retraining a linear classifier every time a new class appears, a prototype classifier does not require such retraining. If a prototype classifier can achieve comparable performance with linear-evaluation methods or meta-learning-based methods, it can be used as a practically useful first step in tackling few-shot learning problems.
>
> ### Q2. More complex feature transformation methods should also have been explored such as exponential, non-linear kernels, power transforms, etc to see whether they can also produce better generalization.
>
> We did not use complex data transformation methods for the following two reasons. The first is that we used data transformation methods based on the generalization bounds derived in the paper, so we do not try such complex transformations that are not explicitly associated with the generalization bounds. The second is that we do not want to use additional hyperparameters as much as possible and want to keep the simplicity of the prototype classifier, so we did not use complex data transformation methods.
>
>
> ### Q3. In Figure 1, don’t the shape of the clusters vary because Euclidean distance is used for ProtoNets while dot-product is used for Linear layers ? Is it because of the loss function ?
>
> As you have pointed out, the difference occurs because dot-product is used for linear layers. We will modify the cross-entropy loss to dot-product with cross-entropy loss.
>
> ### Q4. Can this feature transformation approach improve non-prototypical few shot learning methods ?
>
> We did not perform this experiment in the main paper because we investigated how an easy-to-use prototype classifier would have the same accuracy as a linear classifier or meta-learning-based methods. We conducted additional experiments to see whether the transformations improve the performance of Baseline on miniImagenet.  Table below shows the performance of Baseline with and without the transformation methods.
>
> | | 1-shot        |     5-shot|
> |---|---|---|
> baseline                                                         | 54.54$\pm$0.80 | 76.50$\pm$0.62 |
> baseline w L2-norm                                                   | 53.86$\pm$0.80 | 76.47$\pm$0.65 |
> baseline w EST                                                   | 52.32$\pm$0.83 | 75.56$\pm$0.60 |
> baseline w LDA                                                   | 52.46$\pm$0.80 | 75.62$\pm$0.66 |
> baseline w EST + L2-norm                                               | 54.76$\pm$0.73 | 75.99$\pm$0.68 |
> baseline w LDA + L2-norm                                             | 54.35$\pm$0.72 | 76.93$\pm$0.69 |
>
>
> As shown in the table, the performance does not change so much between the transformation methods. This can be explained as follows. The models are trained with a linear classifier on the features without data transformation and the settings are the same during testing and training. Therefore, the accuracy was sufficiently high and the performance did not change so much when data transformation methods were applied. Furthermore, since our generalization bound is based on a  prototype classifier, not linear classifiers, these results were different from prototype classifiers' results.

---

> > ### Comment · Reviewer_bREF · 2022-08-07
> > **Response to Rebuttal**
> >
> > I would like to thank the reviewers for providing such a thorough feedback. I have increased my score to weak accept.

---

> ### Author Response · Authors · 2022-08-02
> **Response to Reviewer bREF part2**
>
> ### Q5. Can t-SNE feature visualization be shown to see how feature transformation affects discriminability?
>
> We have updated the paper and put the visualization in Appendix A.8.
>
> However, the projection into a lower dimensional space for visualization does not accurately represent the relationship of a higher dimension. Furthermore, the distribution that we can observe qualitatively by visualization is one in which the interclass variance is large and the intraclass variance is small.
> Therefore, we quantitatively evaluated the ratio of the within-class variance to the between-class variance ($\frac{\mathrm{Tr}(\Sigma_\tau )}{\mathrm{Tr}(\Sigma)} $), which is related to our theoretical analysis. We show the results in the following table.
>
> |                                |   miniImageNet | tieredImagenet | CIFAR-FS | FC100 |
> | --- | --- | --- | --- | --- |
> | Baseline w/o transformation  |   4.50             |            3.91         |       4.99       |     10.01 |
> | Baseline w l2-norm      |   4.94                     |    4.31         |        5.18       |     11.36 |
> | Baseline w EST                        |    2.23           |             1.87        |         2.12     |        2.37 |
> | Baseline w EST + L2-norm       |    2.44           |             2.11        |         2.16     |        2.69 |
> | Baseline w LDA                        |    2.15            |             3.37        |         1.99     |       1.64 |
> | Baseline w LDA + l2-norm        |    2.41            |             3.76        |        2.09      |       1.88 |
>
> The Table shows that The L2-normalization  slightly changes the ratio while EST and LDA can reduce the ratio. The combination of EST and L2-norm or LDA and L2-norm can reduce the ratio while decreasing the variance of the norm of feature vectors to 0. This implies that the combination of EST and L2-norm or LDA and L2-norm can reduce the terms expressed in  equation (6) and (8).

---

### Official Review · Reviewer_up5k · 2022-07-12

**Rating:** 6
**Confidence:** 3
**Soundness:** 3 good
**Presentation:** 3 good
**Contribution:** 3 good

**Summary:**

This paper analyzes which factors lead to better performance of a prototype classifier directly applied on top of a learned representation. This stands in contrast to recent work that has shown that training a linear classifier on top is effective but computationally expensive and requires more hyperparameter tuning. Contributions include a new generalization bound on the performance of a prototype classifier that uses fewer assumptions than previous work and better shines light on the critical factors, including variance of the norm, within-class variance, and between-class variance. Experiments are conducted on several benchmark datasets and shows that with appropriate feature transformation and L2 normalization, a prototype classifier can achieve similar performance to a learned linear classifier.


**Questions:**

* As mentioned above, the analysis here is done for a prototype classifier based on Euclidean distance, but cosine distance has also shown to be effective if distances can be scaled appropriately (Oreshkin et al. 2018). To what extent can the current analysis be extended to cosine distance? See also ll. 47-49, where the gap in performance is hypothesized to be due to the different in loss function. Does the gap decrease when cosine distance-based prototype classifiers are used instead?
* The generalization bound presented in Section 3.3 suggests that performance would be improved by controlling the variance of the feature norm. In previous work, hyperspherical prototype networks have been proposed (Mettes et al. 2019), where the norm of feature vectors is enforced to be 1. How does such an approach relate to the  analysis in the paper?
* How do the multiclass results compare to the binary results? Are there any qualitatively different takeaways?

**Limitations:**

More discussion about the limitations of the analysis would be useful: e.g. choice of distance and how this could possibly be addressed.

**Strengths And Weaknesses:**

Strengths
+ The aim is well-motivated. Eliminating the need for retraining a linear classifier and instead simply applying a prototype classifier would make it easier to adapt pre-trained representations to the few-shot setting.
+ The generalization bound removes simplifying assumptions of previous work and in doing so paints a more nuanced picture of the important underlying factors.
+ Experiments are extensive and consist of four datasets and multiple variations of the prototype classifier, including two feature transformation methods both with and without L2 normalization.

Weaknesses
- The clarity could be improved in some places. One is that the discussion of LDA and EST is rather light relative to the important role they play in the experiments. Another is a discussion of how the multiclass bound relate to the binary bound presented in the main paper.
- The discussion of the results could be improved. The important terms are explained at the end of Section 3.3 but it is not very intuitive how future work should go about trying to better control them. I also found Figure 2 somewhat confusing and it was not immediately clear to me how the figure relates to the experimental results.
- The analysis is limited to Euclidean distance, which is a valuable first step, but coupling this with a discussion of e.g. cosine distance would be useful. This is relevant since standard classifiers use inner products to compute logits, which is more closely related to cosine distance than Euclidean.

---

> ### Author Response · Authors · 2022-08-02
> **Response to Reviewer up5k part1**
>
> We would like to thank reviewer up5k for providing valuable feedback and raising interesting questions which we answer below.
>
> ### C1. The discussion of LDA and EST is rather light relative to the important role they play in the experiments.
>
> As shown in figure2 right, LDA and EST can improve the performance of prototype classifiers when the ratio of the within-class variance to the between-class variance of the features is large. However, LDA and EST negligibly improve the performance on datasets with lower within-class variance (miniImagenet, tieredImagenet and CIFAR-FS). This is because the transformation matrix of LDA and EST is estimated on a 5-shot of the testset or the features of the trainset. Therefore, the estimation of the transformation matrix is unstable and thus the performance improvement is relatively small.
>
> ### C2. A discussion of how the multiclass bound relates to the binary bound presented in the main paper is not clear enough.
>
> We extend the conclusion of the bound on binary-classification  to the multiclass case by Frechet’s inequality as shown in Appendix A.5. We will add this at the end of Section 3.3.
>
> ### C3. The important term explained in Section 3.3 is not very intuitive how future work should go about trying to better control them.
>
> From the results in the Remark in Section 3.3 and our experiment, we can observe that reducing the following two terms is important: 1. the variance of the norm of the feature vectors, and 2. the quadratic expression of the ratio of the within-class variance to the between-class variance. Therefore, for example, a transformation that simultaneously reduces the variance of the norm and the ratio of the within-class variance to the between-class variance would be effective. However, using a complex data transformation method would hurt the simplicity of the prototype classifier (which is also the focus of this paper), so we did not try it in our experiment, and conducted the experiment using a combination of simple data transformation methods.
>
> ### C4. Figure 2 is somewhat confusing and it was not immediately clear how the figure relates to the experimental results.
>
> Figure 2 (left) compares the size of each term of our generalization bounds that shown in Section 3.3, and shows that the term of the difference in the shape of the class distribution does not have much impact and thus we should focus on reducing the variance of the norm of the feature vectors and the ratio of the within-class variance to the between-class variance. The right figure shows that FC100 has larger ratio of the within-class variance to the between-class variance compared to the other dataset. Therefore, LDA and EST improve the performance of a prototype classifier on FC100 and only have a small effect on other datasets. The results also show that L2-normalization by itself has little effect on the ratio of the within-class variance to the between-class variance.
>
> ### Q1. The analysis is limited to Euclidean distance, but coupling this with a discussion of e.g. cosine distance would be useful.  To what extent can the current analysis be extended to cosine distance?
>
> The L2-norm is applied after the average operation at the test phase, and this is expressed as
>
>  $$\bar{\phi\left(S_c\right)} = \frac{\frac{1}{K} \Sigma_{x\in S_c} \phi(x)} {|\frac{1}{K} \Sigma_{x\in S_c} \phi(x)|}.$$
>
> The calculation of expectation of $\bar{\phi\left(S_c\right)}$ will be more complicated and thus our analysis cannot be extended to cosine distance straightforwardly.  In a simple way, assuming
>
> $$ \left|\frac{1}{K} \Sigma_{x\in S_c} \phi(x) \right| = 1, $$
>
> our analysis can also be applied in cosine distance.

---

> ### Author Response · Authors · 2022-08-02
> **Response to Reviewer up5k part2**
>
> ### Q2. Does the gap between the performance of a linear classifier and a prototype classifier decrease when cosine distance-based prototype classifiers are used instead?
>
> Yes. The following table shows the performance of ProtoNet, a prototypical network trained with cosine similarity, a prototypical network trained with inner-product and a prototype classifier on the features of Baseline++.
>
> ||1-shot miniImagenet|5-shot miniImagenet|1-shot tieredImagenet |5-shot tieredImagenet
> |---|---|---|---|---|
> |ProtoNet|54.42 $\pm$ 0.86 | 73.56 $\pm$ 0.68 | 56.96 $\pm$ 0.98 | 78.38 $\pm$ 0.71 |
> |ProtoNet w cosine similarity|53.98 $\pm$ 0.83 | 72.38 $\pm$ 0.66 | 53.17 $\pm$ 0.91 | 73.61 $\pm$ 0.77 |
> |ProtoNet w innerproduct|53.87 $\pm$ 0.83 | 71.86 $\pm$ 0.69 | 49.10 $\pm$ 1.00 | 70.11 $\pm$ 0.84 |
> |Baseline@no|46.36 $\pm$ 0.58 | 73.97 $\pm$ 0.62 | 50.60 $\pm$ 0.87 | 78.10 $\pm$ 0.67 |
>
> The performance gap between a prototype classifier based on the features of Baseline and a prototype classifier with cosine distance or inner product is decreased compared to the gap between a prototype classifier based on the features of Baseline and a prototype classifier with Euclidean distance. Note that we l2-normalized only the features of ProtoNet with cosine similarity since the model assumes the norm of the feature vectors to be constant.
>
> ### Q3. How does hyperspherical prototype networks relate to the analysis in the paper?
>
> Hyperspherical prototype networks is different from our analysis in the following point. Since hyperspherical prototype networks prepare prototypes on hypersphere during training the models, it is similar to Baseline++ described in Chen et al. However, We focus on the features of the pre-trained model, not the behavior during training.
>
> ### Q4. How do the multiclass results compare to the binary results?
>
> Since in our experiments and theory, the two-class is a subset of the multiclass, we cannot find any qualitatively different takeaways
>
> ### Q5. More discussion about the limitations of the analysis would be useful: e.g. choice of distance and how this could possibly be addressed.
>
> If we can use the assumptions in the answer of Q1, we can apply our analysis as is for cosine similarity. For other classes of distances, such as Bregman divergence, if the distance equation is expressed with the terms of variance and expected value of the feature vectors, it is possible to expand on this derivation, paying attention to the calculation of the inner product of Bregman divergence.

---

### Official Review · Reviewer_7Qcy · 2022-07-13

**Rating:** 5
**Confidence:** 4
**Soundness:** 3 good
**Presentation:** 3 good
**Contribution:** 3 good

**Summary:**

Recently, the linear classifier on the top of pre-trained model without meta-learning performs comparably to the prototypical networks for few-shot learning. However, these approaches require re-training every time a new class arrives. This paper focuses on prototype classifier which doesn't require re-training or any meta-learning approach. It founds that the variance of the norm of feature vectors affect the performance of prototype classifiers. It then derives a generalization bound that does not depend on assumptions on the class-conditional distributions. Moreover, the paper theoretically shows that this bound decreases as the variance in norm of the feature vectors decreases, and it empirically investigaes different transformation approaches to lower this variance. Finally, the authors analyze the effectiveness of the feature transformations on multiple datasets that include mini-ImageNet, tiered-ImageNet, CIFAR-FS, CUB, and FC-100.

**Questions:**

Please see my questions in the weakness section.

**Limitations:**

Yes, authors adequately addressed the limitations and potential negative societal impact of their work.

**Strengths And Weaknesses:**

### Strengths:

* The paper investigates an interesting problem for few-shot learning by proposing to use prototype classifiers which generally don't require re-training or meta-learning algorithm.

* It theoretically analyze the prototype classifier in terms of variance of the norm of the feature vectors

* It investigates several transformation approaches to minimize this variance that includes L2 normalization, EST,LDA,
EST+L2-norm and LDA+L2-norm

* Experiments on several datasets corresponding to the two different task i.e. i) standard object recognition and ii) cross-domain adaptation.

### Weakness:

* Novelty of generalization bound is limited as it derives a the bound by modifying (Cao et al., 2020)

* 1-shot performance looks good, but for 5-shot experiment, there is negligible performance boost compared to the baseline++ for miniImageNet, tieredImageNet, and CIFAR-FS in Table 1. Could authors provide any explanation or give remarks about it? It is written in the manuscript that all the methods are reimplemented. It could be that the baseline++ was not properly tuned or trained. Even in Table 2, the performance gap is negligible compared to the other methods. Moreover, the same phenomenon can be observed for cross-domain experiments in Table 4 in supplementary materials

* The authors have proposed several feature transformation methods, but there is not a single one which stands out? Can the authors please explain this behavior?

---

> ### Author Response · Authors · 2022-08-02
> **Response to Reviewer 7Qcy**
>
> We would like to thank reviewer 7Qcy for providing valuable feedback and raising interesting questions which we answer below.
>
> ### C1. Novelty of generalization bound is limited as it derives a the bound by modifying (Cao et al., 2020)
>
> As described in Section A.5., our derivation is different from Cao et al.’s study in the following points.
> 1. We re-derived Lemma 3 because the term of the difference between the trace of the class covariance matrices is erased in the lemma. This term cannot be omitted in our derivation since we do not assume the class covariance matrix to be the same among classes.
> 2. We re-derived the bound on the variance of squared Euclidean distance of two vectors, e.g Lemma 4. The derivation of Cao et al. uses the property of quadratic forms of normally distributed random variables and the fact that the sum of normally distributed random variables is also distributed in Gaussian distribution. The calculation of the variance of squared L2-norm without depending on the property of some distributions is not straightforward. We divide the variance of squared Euclidean distance of two vectors into the variance of the norm of the feature vectors and the variance of the inner-product of vectors. Then we apply Cauchy–Schwarz inequality to the inner-product
>
> The conclusion and insights of our bound is also different from Cao’s bound as follows:
> - While Cao's work cannot explain why the L2-normalization of a feature vector can improve the performance of a prototype classifier, our work theoretically shows that the transformation can improve the performance.
> We relax the assumption of Cao’s work; specifically, the bound does not require that the features be distributed in Gaussian distribution, and each covariance matrix does not have to be the same among classes. Therefore, our theoretical results can be applied to the features extracted by a model trained with cross-entropy loss with a linear classifier.
> - While Cao’s bound depends only on the ratio of the within-class variance to the between-class variance, we theoretically show that the bound consists of three terms: (1) the variance of the norm of feature vectors, (2) the difference in the distribution shape constructed from each class embedding, and (3) the ratio of the within-class variance to the between-class variance. We experimentally show that reducing the term (1) and (3) respectively can improve the performance of a prototype classifier.
>
>
> ### Q1. Why is there negligible performance boost compared to the Baseline++ in Table 1 and Table4 for 5-shot settings? . It could be that the Baseline++ was not properly tuned or trained.
>
> Baseline ++ is trained with l2-normalized features while Baseline is not; and thus the performance improvement of  `Baseline ++ w/o L2-norm -> Baseline ++ L2-norm` is larger than `Baseline w/o L2-norm -> Baseline + L2-norm`.
>
> ### C2. The performance gap is negligible compared to the other methods proposed in current studies.
>
> The purpose of this paper is to relax the unrealistic assumptions of existing theories and  make the Prototype classifier more practical and easier to use. Therefore, we aim to show a prototype classifier can perform comparable with other methods in current studies with proper transformation.
>
> ### Q2. Why is there not a single transformation method which performs better than any other methods across all settings?
>
> The reason for this is that it is not the purpose of this paper that we find a  SOTA method in every setting. As we have mentioned in the response to C2, the purpose of this paper is to relax the unrealistic assumptions of existing theories and  make the Prototype classifier more practical and easier to use. Also, since our theory shows that different transformations are needed depending on the nature of the data set. This is the same as the so-called no-free-lunch theorem.

---

### Meta-Review · Area_Chair_qZuR · 2022-08-27

**Recommendation:** Accept
**Confidence:** Certain

**Metareview:**

It has been shown that linear classifier heads on top of pre-trained models can outperform meta-learning approaches. However, this is less adaptable than prototypical classifier heads and requires retraining with each new set of classes. This paper theoretically investigates the generalization of prototypical classifiers and uses this to explore several feature transformations to improve their performance. While there were concerns about the novelty over and above Cao et al., and some minor clarity issues, the reviewers were all generally in agreement that this is a useful contribution to the community and a good starting point for improving prototype classifiers from a theoretical perspective.

**Award:**

No

---

### Decision · Program_Chairs · 2022-09-14

Accept